# TABULAR LEARNING WITH BACKGROUND INFORMATION: LLMS, KNOWLEDGE GRAPHS, OR BOTH?

## ABSTRACT

Tables have their own structure, calling for dedicated tabular learning methods with the right inductive bias. These methods outperform language models. Yet, many tables contain text that refers to real-world entities, and most tabular learning methods ignore the external knowledge that such strings could unlock. Which knowledge-rich representations should tabular learning leverage? While large language models (LLMs) encode implicit factual knowledge, knowledge graphs (KGs) share the relational structure of tables and come with the promise of better-controlled knowledge. Studying tables in the wild, we assemble 105 tabular learning datasets comprising text. We find that knowledge-rich representations, from LLMs or KGs, boost prediction, and combined with simple linear models they markedly outperform strong tabular baselines. Larger LLMs provide greater gains, and refining language models on a KG boosts models slightly. On datasets where all entities are linked to a KG, LLMs and KG models of similar size perform similarly, suggesting that the benefit of LLMs over KGs is to solve the entity linking problem. Our results highlight that external knowledge is a powerful but underused ingredient for advancing tabular learning, with the most promising direction lying in the combination of LLMs and KGs.

## 1 INTRODUCTION: BACKGROUND KNOWLEDGE FOR TABULAR LEARNING

**Tabular learning** Tabular data is central to machine-learning applications, powering applications from healthcare to finance. Yet, tables have properties that set them apart from other modalities. Cells may contain heterogeneous values: numbers, dates, categorical codes, or short texts. These values often only gain meaning through relational context, via column headers and neighboring entries. Tabular learning consists of making row-wise predictions, whether classification or regression, from these heterogeneous features. This unique structure has long favored learning methods with strong inductive biases for mixed-type features, such as gradient-boosted decision trees, over generic deep learning approaches (Grinsztajn et al., 2022; Shwartz-Ziv & Armon, 2022). Recent progress on table foundation models uses transformers with dedicated row-wise architectures, pretrained on synthetic tables (Hollmann et al., 2022; 2025), that are however purely numerical, leaving aside strings, dates, or categories. On the opposite, casting tables to text to readily apply large language models (LLMs) for in-context learning gives excellent few-shot performance, but does not scale nor benefit beyond a few dozen rows (Hegselmann et al., 2023; Gardner et al., 2024).

**Text in tabular learners** Tabular learners, unlike LLMs, leverage the specific repetitions of rows and features for state-of-the-art predictions on tables (Chen & Guestrin, 2016; Hollmann et al., 2025). Yet they also depart from LLMs in that they do not natively model text columns in tables. Here, a particularly underexplored dimension is that these texts often correspond to real-world entities, such as company names, drugs, or locations, that carry latent information far beyond the raw string. Exploiting this background knowledge could substantially improve prediction, especially in small-data regimes where tables themselves do not suffice to infer such knowledge from scratch. For example, a table of clinical trials mentioning drug names could benefit from external knowledge about drug classes, interactions, or approval status. However, state-of-the-art tabular learners are tailored to numbers (Erickson et al., 2025), using pipelines that cast entity strings to opaque numbers: categorical features are one-hot encoded, texts reduced to surface-level representations, e.g., from character n-grams. Doing so discards the opportunity to ground table entries in external knowledge.

How can strings and entities bring background knowledge to tabular learning? A traditional answer would be to use data-integration and database techniques, augmenting tables with features obtained through joins with external databases (Doan et al., 2012; Cappuzzo et al., 2024). Yet, this approach faces well-known obstacles: discovering relevant tables, identifying joins, engineering relevant features while preventing their exponential growth across chained joins (Kanter & Veeramachaneni, 2015). A more scalable alternative enriches tables implicitly by mapping entity strings to vector representations pretrained on large-scale knowledge sources (Cvetkov-Iliev et al., 2023; Grinsztajn et al., 2023; Lefebvre & Varoquaux, 2025). Such embeddings provide compact summaries of factual and relational information from these sources, and can easily be injected into tabular models.

**KGs and LLMs: two opposing philosophies of knowledge**   Pretraining embeddings from knowledge sources is shaped by two opposing philosophies of knowledge.

The *Knowledge Graph (KG) perspective* strives for pure, curated, knowledge. General-purpose KGs (Bollacker et al., 2008; Vrandečić & Krötzsch, 2014; Suchanek et al., 2024) gather facts in a structured, symbolic form, with a high signal-to-noise ratio: what they contain is largely correct. Their strength also lies in their explicit relational modeling, close to the relational nature of tabular data. Yet, their main weakness is their incompleteness: the number of true facts being potentially infinite, no KG can store them exhaustively.

By contrast, the *LLM perspective* treats knowledge as the statistical aggregation of written language. LLMs are probabilistic black-boxes trained on massive, weakly curated text corpora with no explicit notion of truth (Suchanek & Luu, 2023). They do not store curated facts, but model token co-occurrence statistics that implicitly encode fragments of factual knowledge (Petroni et al., 2019; Roberts et al., 2020; Jiang et al., 2020). Their power lies in breadth: scale enables coverage that far exceeds manually constructed KGs. This breadth comes at the price of reliability: LLMs are prone to hallucinations and factual drift (Ji et al., 2023; Tonmoy et al., 2024; Bang et al., 2025; Mallen et al., 2022), may produce confident but incorrect statements (Bender et al., 2021; Kadavath et al., 2022), and their internal reasoning remains opaque (Bender & Koller, 2020; Nanda et al., 2023). Raw application of LLMs to tabular learning also hits the wall of the size of their context window.

**The knowledge integration bottleneck**   While LLMs can easily embed any string, the use of KGs in downstream tasks is hindered by a difficult knowledge integration step. Early KG embedding models operated in a transductive setting, learning representations for a fixed set of entities, primarily for internal tasks like link prediction (Bordes et al., 2013; Yang et al., 2014). Applying these embeddings to external data, such as tables, requires solving the challenging entity linking problem: mapping messy, real-world strings to canonical entities in the KG (Mendes et al., 2011; Delpeuch, 2019; Foppiano & Romary, 2020). This challenge is related to the broader "symbol grounding problem", a central difficulty of symbolic AI (Wikipedia, 2025). Recent advances in KG embedding models strive to overcome this limitation via generalization to unseen entities. One line of work couples KG embedding with pretrained (or jointly trained) text encoders applied to entity names or descriptions, so that unseen entities can be embedded directly from their textual mentions (Wang et al., 2021b; Saxena et al., 2022). A parallel effort focuses on building KG foundation models that can operate in a fully inductive setting, generalizing to entirely new graph structures by reasoning on their topology (Galkin et al., 2023; Huang et al., 2025a). These developments open up new avenues for integrating structured knowledge into downstream applications, but their effectiveness in the context of tabular learning remains an open question.

**Contributions**   We study how to bring background information to tabular learning. Which modality, KGs or open-ended texts, should be preferred to pretrain world-knowledge models? Are numerical table foundation models all we need? What basic components for future research on table foundation models? To answer these questions, we assemble, from three diverse sources with different inclusion biases, 105 tabular learning datasets containing text. We conduct a large-scale empirical study, comparing, in a controlled setting, knowledge-rich representations from both LLMs and KG embedding models of varying sizes. We also study the impact of refining LLMs on KGs, to assess whether this hybrid approach combines the strengths of both modalities. Our findings are threefold:

1. **Bringing knowledge-rich representations into tabular learning matters:** both LLM and KG embeddings improve upon standard encoding techniques such as TF-IDF, bringing more gains on text features than SOTA tabular learners developed for numerical tables.

2. **Entity linking is the key bottleneck:** when all entities in a table are already linked to a KG, LLMs and KG models of comparable size perform similarly, suggesting that the main advantage of LLMs is their ability to implicitly solve the entity linking problem.
3. **Current table foundation models struggle with rich embeddings:** state-of-the-art tabular learners are consistently outperformed by simple linear models on high-dimensional, knowledge-rich representations, revealing a critical limitation.

## 2 RELATED WORK

### 2.1 TABULAR LEARNING WITH TEXT FEATURES

**From tree-based models to foundation models** Historically, tabular learning has been dominated by gradient-boosted decision trees (GBDTs) (Chen & Guestrin, 2016; Ke et al., 2017; Prokhorenkova et al., 2018), which remain strong baselines due to their inductive biases for heterogeneous features (Grinsztajn et al., 2022). Recently, deep learning approaches (Ye et al., 2024; Holzmüller et al., 2024; Gorishniy et al., 2024), including table foundation models pretrained on synthetic data (Hollmann et al., 2022; 2025; Ma et al., 2024; Qu et al., 2025), now markedly outperform GBDTs (Erickson et al., 2025). However, a shared limitation of these methods is that they lack a dedicated mechanism for text features. Instead, they typically rely on simple string preprocessing turning these entries to numerical vectors, and then treat them as any other numerical feature. In practice, this vectorization step often ignores the semantics of string entries, relying on surface-level representations such as TF-IDF or character n-grams that bear no external knowledge.

**Leveraging external knowledge from LLMs and KGs** To address this gap, recent work has explored using external knowledge sources. One prominent approach is to leverage LLMs. Methods like TabLLM (Hegselmann et al., 2023) and Tabula-8B (Gardner et al., 2024) serialize table rows into text and fine-tune an LLM for classification and regression. These works put forward the benefit of in-context learning of LLMs, that brings their excellent few-shot performance to tabular learning, but cannot scale to the size of typical tables. Other work, such as TabStar (Arazi et al., 2025), adapt smaller, efficient text encoders with specialized architectures for tabular data. An alternative paradigm uses KGs as the source of external knowledge. For instance, CARTE (Kim et al., 2024) and TARTE (Kim et al., 2025) pretrain tabular models on KGs, but rely on the simple FastText (Bojanowski et al., 2017) model to process strings.

**Prior comparative studies** A few studies have begun to analyze the benefits of these knowledge-rich representations. Grinsztajn et al. (2023) demonstrated that embeddings from language models outperform traditional substring-based encoders, particularly for columns with diverse text entries. Similarly, Kasneci & Kasneci (2024) showed on 9 datasets that integrating embeddings from models like RoBERTa and GPT-2 into GBDTs often improves performance, especially in low-data regimes. While these works sketch out the value of using language models for text in tables, they do not inform of the relative merits of knowledge sourced from unstructured text (via LLMs) and structured graphs (via KG models).

### 2.2 LEARNING ON KGs

**Structure-based KG models** A long-standing line of research learns representations from KGs by focusing solely on the graph structure. Early models operate in a transductive setting, learning low-dimensional embeddings for a fixed set of entities and relations. Such methods, that include TransE (Bordes et al., 2013), DistMult (Yang et al., 2014), ComplEx (Trouillon et al., 2016), and RotatE (Sun et al., 2019b), model the relations as geometric transformations in the embedding space and define a scoring function to measure the plausibility of triples. To overcome the limitations of transductive learning, subsequent work has focused on inductive models that can generalize to unseen entities (Zhu et al., 2021; Galkin et al., 2021). More recently, this has led to the development of KG foundation models that operate in a fully inductive setting, reasoning on the graph's topology to predict new links on entirely unseen graphs (Galkin et al., 2023; Lee et al., 2023; Huang et al., 2025a;b; Zhang et al., 2025b; Du et al., 2025; Arun et al., 2025). Their application to tables remains however open, as it requires extracting from a table a relational graph rich-enough to enable the inductive setting.

**Text-based KG models**   A parallel approach leverages the textual information associated with entities and relations, such as their names and descriptions. These models typically use a pretrained language model to create text-aware representations, bridging the gap between symbolic knowledge and natural language. One common strategy is to fine-tune a pretrained model such as BERT or RoBERTa using an objective that combines a masked language modeling loss with a KG-specific loss (Wang et al., 2021b;a; Youn & Tagkopoulos, 2022). Other methods frame link prediction as a textual task, either by scoring text sequences representing triples (Yao et al., 2019; Wang et al., 2022b) or by treating it as a sequence-to-sequence problem where the model generates the missing entity's name (Chen et al., 2022; Xie et al., 2022). A prominent example of the latter is KGT5 (Saxena et al., 2022), which verbalizes triples and fine-tunes T5 (Raffel et al., 2020) to predict the missing elements. These text-based approaches enable embedding entities that were not seen during training, a crucial feature for downstream applications.

**LLMs refined on knowledge**   Instead of training a model specifically for KG completion, another line of research refines general-purpose LLMs with structured knowledge to enhance their factual grounding. This approach aims to inject the high-quality, curated facts from KGs into the broader world knowledge implicitly stored in LLMs. For example, the ERNIE line of work (Sun et al., 2019a; 2020; 2021) refines language models like RoBERTa (Liu et al., 2019) by incorporating knowledge-base data into their pretraining objectives. More recently, the Knowledge Card framework (Feng et al., 2023) demonstrated that fine-tuning a moderately-sized LLM such as OPT-1.3B (Zhang et al., 2022) on KG triples can effectively plug factual knowledge into larger LLMs, improving their performance on knowledge-intensive tasks. Retrieval-based methods (Lewis et al., 2020) offer a complementary paradigm, dynamically fetching knowledge at inference time rather than encoding it statically, and represent a promising alternative for knowledge integration.

## 3   METHODOLOGY: A BENCHMARK FOR TABLE BACKGROUND KNOWLEDGE

### 3.1   105 TABULAR DATASETS

**Three diverse data sources**   To ensure the robustness and generality of our findings, we assemble a diverse benchmark of 105 tabular datasets from three sources with distinct characteristics and inclusion biases: TextTabBench (Mráz et al., 2025), CARTE (Kim et al., 2024), and WikiDBs (Vogel et al., 2024).

Table 1: Task distribution across sources.

| Source | b-clf | m-clf | reg | Total |
|---|---|---|---|---|
| TextTabBench | 5 | 2 | 10 | **17** |
| CARTE | 11 | 0 | 40 | **51** |
| WikiDBs | 1 | 21 | 15 | **37** |
| **Total** | **17** | **23** | **65** | **105** |

TextTabBench and CARTE are established benchmarks for tabular learning, providing real-world tables with varied text features, from short entity names to longer descriptions. Each table is associated with a predefined prediction task (regression, binary, or multi-class classification). WikiDBs is a large corpus of over 1.6 million semi-synthetic tables generated from Wikidata. To create meaningful tasks from this source, we first filtered for tables with

Table 2: Aggregated features of tabular datasets across sources. The cardinality is computed on 1,024 rows.

| | **TextTabBench** | **CARTE** | **WikiDBs** |
|---|---|---|---|
| # columns | 15.65 | 6.76 | 6.73 |
| cardinality | 286.36 | 371.44 | 463.70 |
| string length | 975.29 | 298.80 | 203.62 |
| string similarity[1] | 0.16 | 0.10 | 0.08 |

[1] cosine similarity of TF-IDF across rows

at least 1,200 rows, then manually curated a subset of 37 tables for which we could define a relevant prediction problem. Table 1 summarizes the final distribution of tasks across the three sources. Further details on each dataset are available in the Appendix (Table 8, Table 9, Table 10).

**Data preprocessing**   We adopt the original preprocessing from TextTabBench and CARTE. For WikiDBs, we apply a procedure similar to TextTabBench. We also ensure that multi-class classification tasks have at most 10 classes, each with at least 105 samples. For all 105 datasets, we then apply the following preprocessing pipeline: (1) we remove all numerical columns to focus our study on text-based knowledge (expect in subsection 5.2); (2) we log-transform regression targets with

Figure 1: An overview of our evaluation pipeline. For each dataset, we sample training and test sets. We then serialize the rows and use the embedding **model** to generate a vector representation for each row. Finally, we train a tabular learning estimator to evaluate these embeddings.

wide-ranging distributions; (3) we downsample majority classes in multi-class problems to create balanced datasets; and (4) we discard any table with fewer than 1,050 rows post-processing to ensure sufficient data for evaluation. We also exclude one dataset from TextTabBench with excessively long text entries that exceed the context limits of some of our baselines.

**Linked tables for controlled comparison** To isolate the contribution of knowledge from the challenge of entity linking, we create a specialized subset of 15 tables where text entries are unambiguously linked to entities in Wikidata5M (Wang et al., 2021b), a large-scale KG derived from Wikidata. These tables are selected from our main benchmark if they contain at least 1,050 rows with entities that can be matched to the KG. For this subset, we retain only the entity column and the prediction target, and remove all unlinked rows. This setup allows for a direct comparison of pure KG models with LLMs in a scenario where entity linking is solved.

Table 3: Knowledge graph datasets. Smaller versions of Wikidata5M are created by filtering entities by degree ("deg."). All graphs use the same 822 relations.

|  | # entities | # triples | deg. |
|---|---|---|---|
| Wikidata5M | 4.6M | 20.6M | - |
| Wikidata3M | 3.2M | 15.5M | 3 |
| Wikidata2M | 2.1M | 11.5M | 4 |
| Wikidata1M | 1.1M | 6.8M | 6 |
| Wikidata500k | 0.5M | 3.1M | 9 |

To analyze the impact of KG size, we generate four smaller KGs by progressively filtering out low-degree entities and retaining the largest connected component of the induced subgraph. The statistics of these graphs are presented in Table 3.

## 3.2 EVALUATION PIPELINE

Our evaluation pipeline, summarized in Figure 1, assesses the quality of representations from various knowledge sources for downstream tabular tasks. For each dataset, we generate row-wise embeddings from a given model and then train a tabular predictor to predict the target variable from them.

**Experimental setup** To simulate small-data scenarios where external knowledge is most critical, we sample training sets of varying sizes, $n_{\text{train}} \in \{64, 256, 1024\}$. The test set consists of 1,024 held-out samples (or all remaining samples if fewer are available). To ensure robust evaluation, we repeat this process 10 times with different random seeds for each configuration.

**Embedding models** We evaluate a wide range of models to generate representations:

- **Non-pretrained baseline:** As a simple baseline without external knowledge, we use a TF-IDF vectorizer followed by a Truncated SVD with 30 components per column, implemented in the `Skrub` library.

- **Pure LLMs:** To study the effect of model scale and architecture, we include a diverse set of pretrained language models: the Llama-3.1 family (1B, 3B, 8B) (Dubey et al., 2024), the Qwen3 Embedding series (0.6B to 8B) (Zhang et al., 2025a) which performs well on the Massive Text Embedding Benchmark (Muennighoff et al., 2022), RoBERTa (base, large), T5 (small, base), e5-v2 (small, base) (Wang et al., 2022a), and OPT-1.3B. We also include FastText as a representative of shallow, non-transformer text models.
- **Hybrid LLM+KG models:** To assess the benefit of structured knowledge, we evaluate models that refine LLMs on relational data. This includes ERNIE 2.0, KGT5, Knowledge Card, Tabula-8B, and TARTE. Each model is compared against its corresponding LLM.
- **Pure KG models:** For the subset of 15 linked tables, we evaluate classic KG embedding models: DistMult, TransE, ComplEx, and RotatE. We train these models on Wikidata5M and its subsets, using an embedding dimension of $d = 300$.

**Table serialization and downstream estimators**  To generate embeddings from LLMs, we serialize each row into a natural language prompt. Following Gardner et al. (2024), we use the format: "The <col_a> is <val_a>. The <col_b> is <val_b>. What is the value of <target>?". For KGT5, we adapt the prompt to better match its pretraining format: "<col_a> | <val_a>. <col_b> | <val_b>. Predict: <target>". Constructing the embeddings across multiple columns (as opposed to Grinsztajn et al. (2023)) enables the context (column name, other entries on the same row) to inform the representation, e.g. leading to disambiguate "Cambridge; UK" from "Cambridge; Massachusetts" in a table with columns "city; country".

The resulting high-dimensional embeddings are then fed into three representative tabular learners:

- **Ridge regression:** A simple and efficient linear model.
- **XGBoost:** A powerful GBDT model. To manage computational cost, we first reduce the embedding dimensionality to 300 using PCA. We then perform hyperparameter optimization via a randomized search (see Table 6).
- **TabPFNv2:** A transformer-based table foundation model, doing in-context learning. We use PCA to reduce dimensionality to 500, the maximum supported by the model.

## 4  RESULTS: KNOWLEDGE REPRESENTATIONS FOR TABULAR LEARNING

### 4.1  KNOWLEDGE-RICH REPRESENTATIONS BOOST TABULAR LEARNING

**More gains from knowledge representations than advanced tabular learning**  Figure 2 shows that, for text features, improving the quality of the representations leads to more gains than using advanced tabular learning methods. Indeed, the best performance across the 105 datasets is obtained by a simple predictor, Ridge, applied on good representations, such as those created via modern LLMs, outperforming sophisticated tabular learning methods XGBoost and TabPFNv2 (Figure 2a). In addition, more sophisticated tabular-learning models benefit less from advanced representations. This could be either because their flexibility enables them to fill-in for a less rich representation, or because the representations do not match their implicit inductive biases, tailored for tabular learning. Indeed, unlike typical tabular data, these representations are high-dimensional and closer to being rotationally-invariant Grinsztajn et al. (2022). Moreover, these advanced tabular learners cannot be applied as such to the knowledge-rich representations, as they have too many features. Thus we need to reduce the input dimensions with PCA (see subsection 5.1), following Grinsztajn et al. (2023).

A complementary observation is that the benefit of adding knowledge-rich representations to a simple tabular learner is larger than the benefit of using a sophisticated tabular learner on simpler representations: Figure 2b shows that TabPFNv2 achieves only half of the performance gains of Ridge combined with a good LLM-based representation.

**Benefits for a wide variety of tables, from multiple sources**  Figure 2b shows that, for the Ridge learner, knowledge-rich representations bring an improvement over non-pretrained string representation across methods, and larger models benefit consistently across the three different sources (Figure 16 gives source-specific results). These datasets are varied (Table 2), and the different sources represent different selections of tables with text. This diversity suggests that knowledge-rich representations help tabular learning in general, when the tables have text columns. The benefit is, on

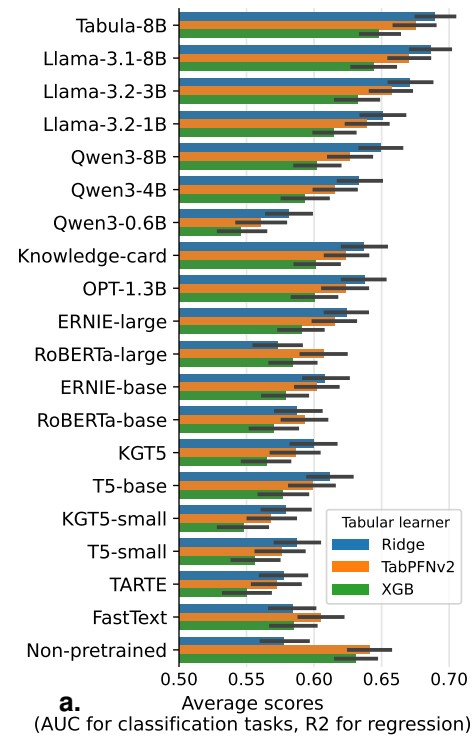 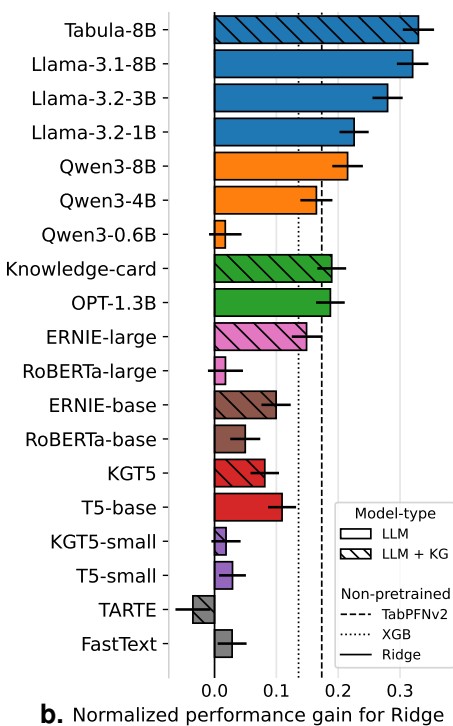

Figure 2: Performance gain of the various knowledge-rich representations compared to a non pretrained baseline – **a.** Comparisons including three tabular learners: Ridge, XGBoost, and TabPFNv2; absolute scores. – **b.** Relative improvements to non-pretrained string representations, when using a Ridge model as a tabular learner; normalized scores (0 is 10% worse, 1 is best score observed). – Appendix Figure 15 gives critical difference diagrams across all methods and datasets.

average, quite marked: going from non-pretrained string representation to the best LLM-based ones gives a .2 average boost in AUC or R2 to Ridge (though only a .05 boost to TabPFNv2).

## 4.2 LARGER LLMs BRING MORE VALUE

Figure 3 shows the performance gain as a function of the LLM size (number of parameters), focusing only on pure LLM representations. It reveals that the benefit increases as a function of size, for transformer-based representations (thus excluding FastText, which is a big model but very wide and shallow). This benefit of size is very clear in a given model family (comparing various sizes of e5, Qwen, or Llama-3). We hypothesize that this general scaling is driven by larger representational capacities brought by the increased number of parameters that enables the storage of more prior knowledge.

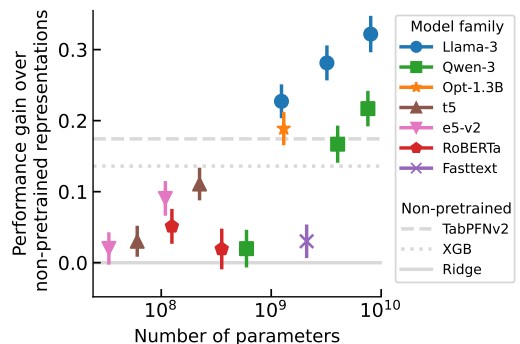

Figure 3: Effect of the size of the model, for pure-LLM representations.

## 4.3 REFINING LLMs ON KGs BOOSTS LANGUAGE MODELS SLIGHTLY

Figure 4 compares the benefit brought by each method that has refined an LLM on a knowledge graph or knowledge base to the corresponding non-refined base LLM, as a function of size.

We estimate the scaling of the performance as a function of the number of parameters with a linear regression for both families of approaches –LLMs with and without KG refinement. Both families show the same scaling, but refining on KGs brings an offset: it enables reaching the same performance with a model with a number of parameters smaller by a factor of 2/3rd.

Note that the data points with the largest model correspond to the pair Tabula/Llama3 (Gardner et al., 2024), which refines on tabular data rather than a rich KG. This pair also displays a comparatively smaller benefit of the refinement, which may result from the limited richness of the corresponding data.

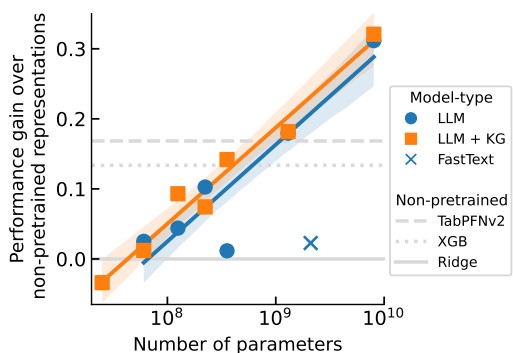

Figure 4: Comparison of LLM and their matched counterpart refined on knowledge bases.

The observed benefit of refining LLMs on KGs raises the question: what do knowledge graphs add to LLMs? How important is a rich knowledge graph?

### 4.4 TEASING OUT KNOWLEDGE FROM ENTITY MATCHING: TESTING PURE KG SOLUTIONS

**Under automatic and noisy entity linking** To compare LLMs with pure KG models, we use BLINK (Wu et al., 2020) for automatic entity linking (subsection A.4), allowing us to incorporate KG embeddings. On 33 datasets where text entries are linked to Wikidata5M, LLMs consistently outperform KG embeddings (Figure 5). However, the noisy entity linking confounds the comparison, making it unclear if the performance gap primarily comes from better knowledge representation or from linking failures.

Indeed, LLMs are more than pure knowledge engineering objects: applied to embed texts, as we do here, they also bring in a form of fuzzy matching of entities (technically related to recontextualizing the tokens) and language understanding. This is to be contrasted with KGs, which are pure knowledge engineering objects (arguably with crisper knowledge), but 1) require entity matching and 2) do not bring language understanding.

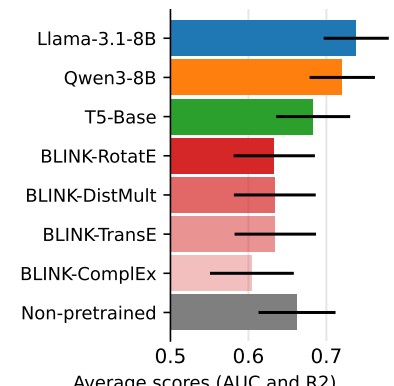

Figure 5: Comparing LLMs to KG embeddings after automatic entity linking with BLINK.

**Idealistic setting: perfect entity linking** To tease out the role of background knowledge, we investigate a subset of tables for which the entity matching problem is solved, and each entry is linked to an entity in Wikidata5M.

In such an ideal scenario, pure KG embedding approaches provide features for the tables entries (Grover & Leskovec, 2016; Cvetkov-Iliev et al., 2023; Robinson et al., 2024). Figure 6 compares the benefits of LLM-based approaches with KG embedding approaches, varying the size of the models. For KG embedding, the size of the model is varied by varying the size of the KG used to build the embeddings (see Table 3): a smaller KG represents fewer entities, and thus has fewer parameters. When we reduce the size of the KG, it only provides representations for a fraction of the entities of the downstream table, and thus the downstream performance. This decrease is sharper than for LLMs,

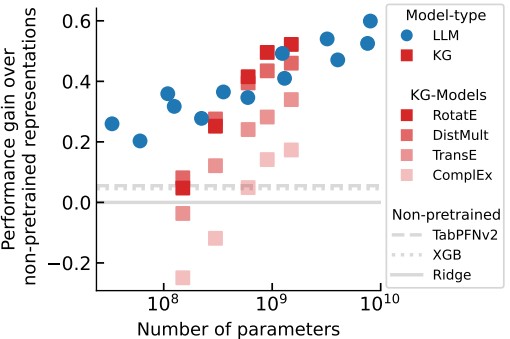

Figure 6: Comparing pure KG to pure LLM approaches on perfectly matched tables.

because smaller KGs face a hard failure (entity is not matched) while language models face a soft failure: they give an embedding whatever the query is. This embedding can be of varying quality, sometimes extrapolating beyond the knowledge of the LLMs, which corresponds to hallucination. However, an extrapolation that is only partly correct can still help downstream tabular learning.

**Without entity-matching challenges, KG embedding is on par with LLMs** For the largest, non-reduced KG, all table entries are matched, and good KG embedding models perform as well as LLMs of the same size (Figure 6). Interestingly, this suggests that for the same number of parameters, KG embeddings do not store crispier knowledge than LLMs.

**Driven by knowledge, rather than language understanding** On the converse, when all entities are matched to the KG, similarly-sized LLMs bring no benefit. This suggests that their language-understanding features are not critical for these tasks. However, the selection of tables with entities all represented in KGs may introduce a bias towards more knowledge-centric tasks.

## 5 ABLATION STUDIES: PCA AND NUMERICAL FEATURES

### 5.1 STUDY OF THE IMPACT OF PCA

**Is Ridge outperforming XGBoost and TabPFNv2 because of PCA?** To determine whether the lower performance of XGBoost and TabPFNv2 stems from dimensionality reduction or from the estimators themselves, we evaluate Ridge regression on PCA-reduced embeddings. This ensures a controlled comparison, since all three estimators (Ridge, XGBoost, TabPFNv2) share identical input vectors. Figure 7 presents the results with PCA dimension $d = 300$. We observe that Ridge still outperforms XGBoost and TabPFNv2 even when restricted to the same reduced inputs. This suggests that the performance gap is not an artifact of PCA, but rather reflects the inability of these tabular learners to fully leverage the embeddings.

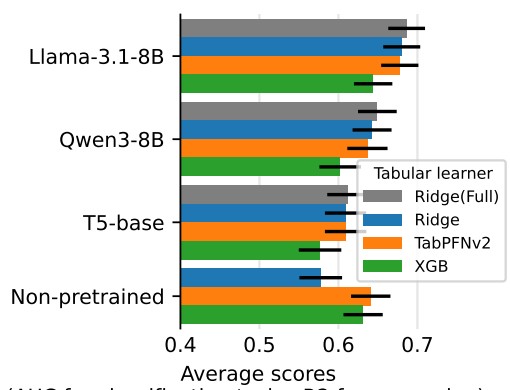

Figure 7: Comparing downstream estimators on the same PCA-reduced embeddings ($d = 300$). Grey bars represent Ridge without PCA. For non-pretrained representations, there is no PCA.

**Does PCA hurt performance?** To assess the impact of dimensionality reduction, we compare the performance on the original embeddings versus PCA-reduced versions of different sizes. Figure 8a shows that PCA incurs only a small performance drop for Ridge. However, for TabPFNv2, decreasing the input dimensionality from 500 to 300 surprisingly improves performance (Figure 8b). This shows that TabPFNv2 struggles with high-dimensional inputs, hindering its ability to leverage rich

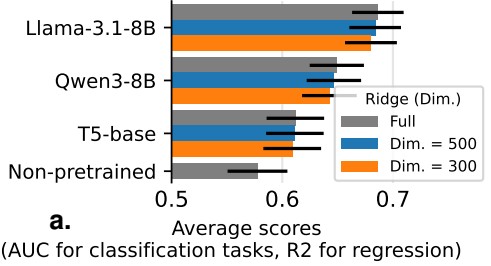
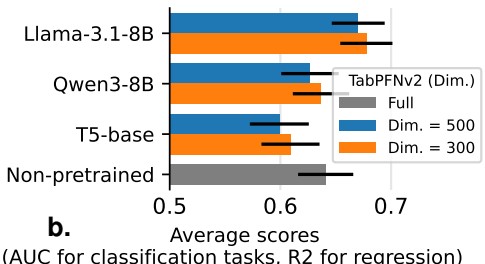

Figure 8: Effect of PCA on performance. **a.** Comparing Ridge with and without PCA. The grey bar represents the performance of Ridge without PCA. **b.** Comparing TabPFNv2 on PCA-reduced embeddings with $d = 300$ and $d = 500$.

embeddings. This limitation may stem from challenges that large contexts pose to transformers. By providing a more compact representation, PCA ultimately aids TabPFNv2, despite information loss.

## 5.2 RE-INTRODUCING NUMERICAL FEATURES

To assess how findings generalize to tables with mixed data types, we reintroduce numerical features and evaluate performance on three settings: text-only, numerical-only, and combined. Our key observations hold (Figure 9). First, combining numerical and textual features markedly outperforms using either modality alone, demonstrating that they bring complementary information. Second, text-only features are more predictive than numerical-only features on these datasets, underscoring the importance of text representation. Third, the relative ranking of text encoders remains consistent when numerical features are included. Finally, while knowledge-rich representations bring substantial gains to a simple linear model like Ridge, they offer only marginal improvements for a table foundation model like TabPFNv2, highlighting its difficulty in leveraging high-dimensional, knowledge-rich embeddings.

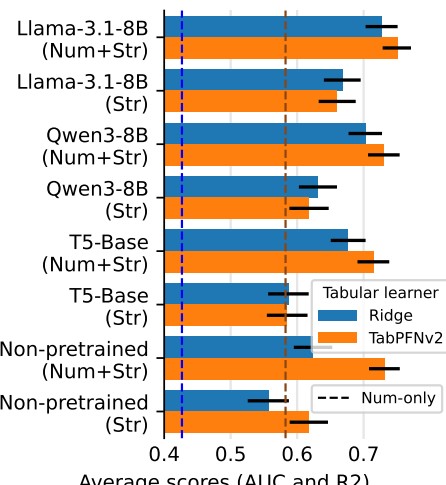

Figure 9: Average performance when using text-only (Str), numerical-only (dashed lines), or combined (Num+Str) features.

## 6 DISCUSSION AND CONCLUSION

**External knowledge is a powerful, yet underleveraged, ingredient for tabular learning** Our large-scale study demonstrates that representations from knowledge sources, whether LLMs or KGs, consistently improve prediction over standard text encodings, and bring predictive information complementary to numerical features. For text features, gains from using better representations with a simple linear model surpass those from applying state-of-the-art tabular learners on less informative representations, suggesting that for tables with text, the primary bottleneck lies in representing text well, rather than in the tabular learning algorithm. Research is needed on privacy and robustness for these new settings, as external knowledge may introduce side channels or adversarial attacks.

**LLMs solve symbol grounding, KGs provide curated knowledge** Direct applications of KG embeddings are hindered by the difficult entity linking step. Using automatic linking solutions incurs substantial computational costs, and results in lower performance than LLMs, which are directly applicable to any text. Yet, when entities are pre-linked, KG embeddings match LLMs of similar size. This implies that the main advantage of LLMs is not superior knowledge, but rather their ability to solve the symbol grounding problem. Our findings point to a promising synergy: refining LLMs on KGs improves performance, making models more parameter-efficient, with refined models achieving the performance of pure LLMs roughly 1.5 times their size, while whether dynamic, retrieval-based approaches could further boost performance remains an open question.

**Current table foundation models struggle with rich representations** While state-of-the-art table foundation models like TabPFNv2 excel with numerical features, they falter when facing high-dimensional embeddings. On these rich inputs, they are consistently outperformed by simple linear models. More strikingly, their performance improves when the embeddings are further compressed via PCA, revealing a core inability to process rich, high-dimensional information. As text is a key component of many tables, future work should develop architectures that can effectively integrate both rich textual representations and numerical features to realize their combined predictive power.

**Scaling up: larger LLMs and broader knowledge** Our results highlight the critical role of scale, yet current tabular methods rely on small language models (Kim et al., 2024; 2025; Arazi et al., 2025). Future foundation models should leverage larger LLMs combined with massive knowledge bases. Resources like Wikidata, with over 100M entities, remain largely underexploited, representing a major opportunity for pretraining powerful, knowledge-grounded tabular learners.

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

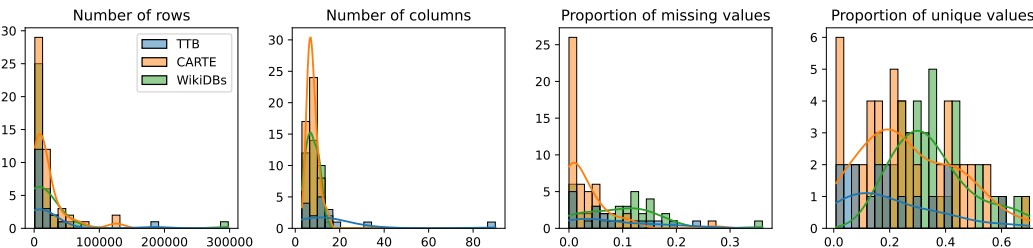

Figure 10: Statistics distribution across sources.

# A  MORE DETAILS ON THE EXPERIMENTS

## A.1  DATASETS

**More statistics on datasets**  Figure 10 gives statistics about table sizes, proportion of missing values, and mean column cardinality.

Table 8, Table 9 and Table 10 provide details on each individual dataset.

**Experiments on linked tables**  We have 15 linked tables, 4 for classification and 11 for regression. Details on these tables are provided in Table 4.

Table 4: Task distribution across sources, for linked tables.

| Source | b-clf | m-clf | reg | Total |
|--------|-------|-------|-----|-------|
| CARTE | 0 | 0 | 3 | **3** |
| WikiDBs | 1 | 3 | 8 | **12** |
| **Total** | **1** | **3** | **11** | **15** |

## A.2  MODELS

**Extracting embeddings from LLMs**  We generated sentence-level embeddings from the serialized rows using the `SentenceTransformer` framework (Reimers & Gurevych, 2019), which provides a unified interface for a wide range of transformer-based models. We used it to extract representations from the following models: Llama, Qwen, RoBERTa, T5, e5-v2, OPT, Tabula, ERNIE, Knowledge Card, and KGT5 families, using pretrained checkpoints available on the Hugging Face Hub (Wolf et al., 2020).

**Embedding dimensions**  Table 5 reports the embedding dimensions for the different baseline models used.

**Incorporating KG embeddings in tables**  For the KG embedding models (DistMult, TransE, ComplEx and RotatE), we use $d = 300$ for the embedding dimension, and train them for 100 epochs with a batch size of 8192 and a learning rate of $10^{-3}$, and use the default parameters of their PyKEEN implementation (Ali et al., 2021).

Table 5: Embedding dimensions for the different baseline models.

| Model | Dimension |
|-------|-----------|
| TF-IDF + SVD | 30 per column |
| FastText | 300 |
| TARTE | 768 |
| Llama-3.2-1B | 2048 |
| Llama-3.2-3B | 3072 |
| Llama-3.1-8B | 4096 |
| TabuLa-8B | 4096 |
| Qwen3-0.6B | 1024 |
| Qwen3-4B | 2560 |
| Qwen3-8B | 4096 |
| RoBERTa (base, large) | 768, 1024 |
| ERNIE 2.0 (base, large) | 768, 1024 |
| e5-v2 (small, base) | 384, 768 |
| T5 (small, base) | 512, 768 |
| KGT5 (small, base) | 512, 768 |
| OPT-1.3B | 2048 |
| Knowledge-card | 2048 |

For KGs smaller than Wikidata5M (see Table 3), some rows of the linked tables are not matched to the KG. In that case, after embedding the rows corresponding to matched entities, we impute missing values using the mean along each column. If no row at all is matched in a table, we simply replace the missing values with zeros.

**XGBoost hyperparameter tuning** For the XGBoost estimator, we perform hyperparameter optimization via a randomized search with 100 iterations. We use 5-fold cross-validation, repeated 5 times on the training set, to evaluate each hyperparameter configuration. The detailed search space is provided in Table 6.

Table 6: Search space for XGBoost hyperparameters.

| Hyperparameter | Distribution | Range |
|---|---|---|
| n_estimators | Integer | [50, 1000] |
| max_depth | Integer | [2, 6] |
| min_child_weight | Log-uniform | [1, 100] |
| subsample | Uniform | [0.5, 1.0] |
| learning_rate | Log-uniform | $[10^{-5}, 1]$ |
| colsample_bylevel | Uniform | [0.5, 1.0] |
| colsample_bytree | Uniform | [0.5, 1.0] |
| gamma | Log-uniform | $[10^{-8}, 7]$ |
| reg_lambda | Log-uniform | [1, 4] |
| alpha | Log-uniform | $[10^{-8}, 100]$ |

### A.3 RESULT REPORTING

**Metrics and score normalization** We evaluate performance using the $R2$ score for regression and the ROC-AUC score for classification. To aggregate results across datasets of varying difficulty, we normalize scores for each dataset and random seed. Following Grinsztajn et al. (2022), we establish a normalized scale where the best-performing model scores 1 and the model at the 10th performance percentile scores 0. Other models' scores are mapped to this [0, 1] range via an affine transformation. For regression, we clip scores at 0 to mitigate the impact of poor-performing outliers.

However, to showcase real effect sizes, we also report original, non-normalized scores in some figures. Figures with non-normalized results are labeled with metric names ("AUC and R2"), while those with normalization are labeled "normalized score".

**Uncertainty estimation** To account for statistical variability, we repeat each experiment 10 times with different random seeds. The error bars in our result figures represent the standard error of the mean across these runs.

### A.4 USING BLINK FOR AUTOMATIC ENTITY LINKING

BLINK (Wu et al., 2020) is a BERT-based entity linking tool that matches entity mentions within texts to Wikipedia entities. It uses a bi-encoder to retrieve candidates by embedding mention contexts and entity descriptions, and a cross-encoder to re-rank them.

Since BLINK requires a textual context (left context, mention, and right context) not natively present in tables, we need to transform tabular data into the required input format. To do so, we implement the following pipeline:

1. **Column selection:** We first manually identify the columns in each dataset in which we expect to find Wikipedia entity mentions to be linked. For instance in the Fifa22 Players dataset (Table 12), we exclude the work_rate and body_type columns.
2. **Context generation:** Each table row is converted into a sentence using the template: "The dataset is <dataset_name>. The <col_a> is <val_a>. The <col_b> is <val_b>. ...". Compared to the serialization of our main study, we add the dataset name, and remove the target name.
3. **Applying BLINK:** For each value in the selected columns, we treat the value as the "mention" and the rest of the generated sentence as its context. We then use BLINK to retrieve the top two Wikipedia entity candidates.
4. **Filtering matches:** To improve linking-quality, we discard the candidites for which the model is not confident. Specifically, we consider a match successful only if the score of the top candidate is greater than the second candidate's score by a margin of at least 1, indicating high confidence.
5. **Mapping and embedding:** We map the successfully linked Wikipedia entities to their Wikidata5M counterparts using mapping files from Wikimedia[2]. The linked columns are then represented by their corresponding KG embeddings pre-computed on Wikidata5M. For all other text columns, we use a non-pretrained TF-IDF + SVD representation.

---

[2]https://dumps.wikimedia.org/enwiki/latest/

6. **Prediction:** Finally, we concatenate the embeddings from all columns and use them as input for Ridge. We report the results in Figure 5.

To manage the computational cost of BLINK, we conducted this experiment only on a subset of 33 tables, each containing fewer than 10,000 rows. The entire process for these datasets took approximately 16 hours. Table 7 provides further details, including the specific columns selected for linking, the proportion of entries for which a match was found, and the runtime of BLINK for each dataset.

# B    ADDITIONAL ANALYSIS

## B.1    STUDYING THE EFFECT OF TRAIN SIZES BEYOND 1,024 SAMPLES

To broaden the scope of our study, we extend our analysis to larger train sizes. On a subset of 49 datasets with more than 10,000 rows (8 from Text-TabBench, 27 from CARTE, and 14 from WikiDBs; see Table 8, Table 9, Table 10) we plot learning curves ranging from 64 to 10,000 samples, for several representative models. As shown for Ridge in Figure 11, the benefits of knowledge-rich representations persist as the training size increases. While larger training sets improve performance for all models, their relative ranking remains largely unchanged.

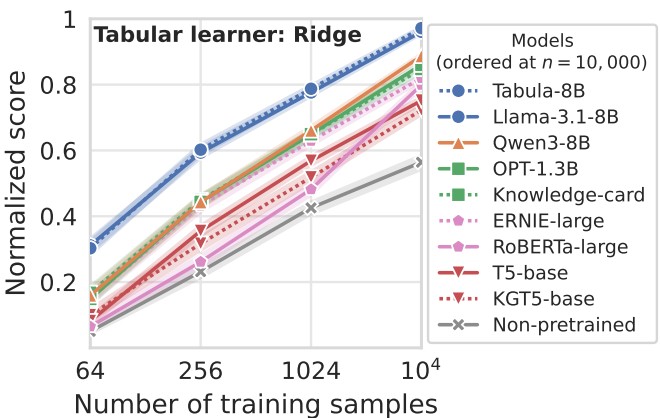

Figure 11: Learning curves on datasets with more than 10,000 rows. The results are shown for Ridge.

## B.2    DOES COLUMN CONTEXT ACTUALLY BRING VALUE?

To study the importance of column context in our pipeline, we run the experiments with a different serialization that does not incorporate the column names. Specifically, each row is now serialized into the following sentence: "The value is `<val_a>`. The value is `<val_b>`. What is the value of target?".

Figure 12 compares the performance with and without column context, for a few representative text encoding models (LLaMA-3.1-8B, Qwen3-8B, and T5-base). We see that on average, the effect of adding column context is positive, but small. However, the $p$-values of a one-sided $t$-test show that for both TabPFNv2 and Ridge, this effect is statistically significant. Interestingly, we also see that, while for most datasets adding the column context helps, for some others it deteriorates the performance.

To better understand these results, we conduct a qualitative study of these datasets in subsection C.2.

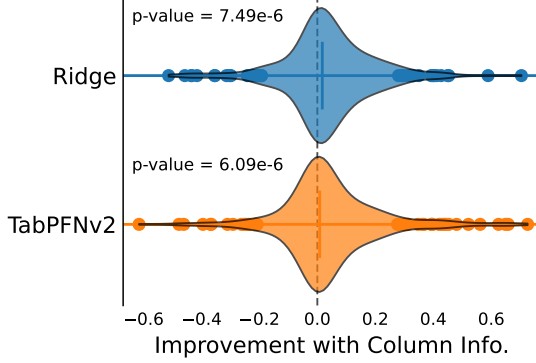

Figure 12: Impact of column context in serialization. The violin plots show the distribution of the difference in normalized scores (with context vs. without). Positive values indicate that adding column context improved performance. Outlier datasets are represented with dots, and the mean with a vertical bar. P-values of a one-sided t-test are reported for Ridge and TabPFNv2.

### B.3 EFFECT OF MODEL SIZE ON TABLES WITH NUMBERS

In Figure 13, we analyze the effect of model size on datasets that contain both numerical and textual features. We observe a clear and consistent scaling trend: larger models within each class outperform smaller ones. Interestingly, incorporating numerical features alongside text embeddings yields similar improvements across all text encoders, suggesting that the information captured by richer models is complementary to, rather than redundant with, numerical features.

## C QUALITATIVE EXAMPLES

### C.1 EXAMPLES FOR LLMS VS KGS

To illustrate the distinct advantages of pure LLMs versus KG-refined models, we examine two representative datasets from our benchmark.

Table 11 displays a snippet from the `Customer Complaints` dataset, where pure LLMs perform very well. The table includes columns with free-form text, such as the `Issue` column. LLMs, trained on open-ended text, are well-suited to process such unstructured language. In contrast, models refined on KGs, like KGT5, may struggle as their pretraining focuses on structured facts and short canonical entity names, making them less suited for open text.

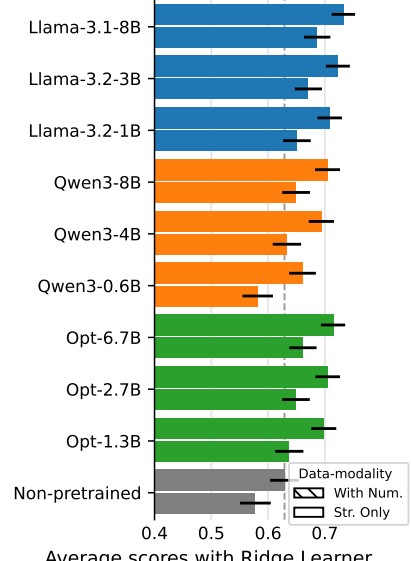

Figure 13: Effect of model size when using both numerical and textual features. The results are shown for Ridge.

Conversely, Table 12 presents an excerpt from the `Fifa22 Players` dataset, where KG-refined models demonstrate strong performance. The task is to predict a player's wage, which is highly knowledge-intensive. By injecting structured factual knowledge during pretraining, KG-refined models gain an advantage for such tasks, leveraging external information to make more accurate predictions.

### C.2 EXAMPLES FOR COLUMN CONTEXT

We present two datasets to illustrate the impact of column context in the serialization.

First, in the `Registered Ships` dataset (Table 13), adding column context is beneficial. Informative headers like `ShipName` and `Shipbuilder` provide crucial information about the type of data in each column, and help the model disambiguate entities. For instance, the string "Otto Hahn" alone typically refers to the German chemist, but when prefixed with `ShipName`, it can be correctly identified as a ship.

Conversely, in the `Company Employees` dataset (Table 14), column context degrades performance. Here, generic column names such as `name` and `domain` do not provide valuable additional information. Including them in the serialization may distract the model from the more informative cell content, leading to a drop in performance.

## D ADDITIONAL RESULTS

### D.1 RUNTIME ANALYSIS

The benefits of leveraging external knowledge come at a computational cost. Table 15 details the average runtimes for embedding generation and estimator fitting (Ridge) across different embedding models and training sizes. As expected, larger models introduce a significant computational overhead. For instance, generating embeddings with an 8-billion-parameter LLM is, on average,

over 100 times slower than using the non-pretrained baseline. This highlights the trade-off between predictive performance and the computational resources required for knowledge integration.

## D.2 RAW RESULTS

Table 16, Table 17 and Table 18 provide the raw results for regression and classification on Text-TabBench, CARTE and WikiDBs datasets respectively, aggregated over 10 random seeds and for train size $n_{\text{train}} = 1,024$.

## D.3 COMPARISON WITH TARTE-FT

To benchmark our modular approach, combining knowledge-rich representations with downstream learners, against end-to-end baselines capable of jointly modeling heterogeneous data (strings and numbers), we evaluate TARTE-FT (Kim et al., 2025) on the 51 CARTE datasets, using both numerical and text features. TARTE is pretrained on a large knowledge corpus derived from Wikidata, and can operate either as (i) a frozen table featurizer (as in our main experiments) or (ii) a fine-tuned model on a specific downstream task (TARTE-FT), for enhanced performance. Because TARTE was originally developed

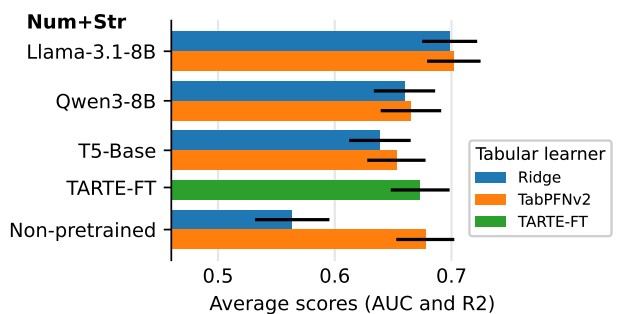

Figure 14: Comparison of TARTE-FT with modular approaches combining pretrained representations and downstream estimators. Results are shown on the CARTE datasets, using both textual and numerical features.

for mixed tables, its weaker performance in our string-only experiment (Figure 2) could be expected. Its base text encoder is FastText, whereas our strongest baselines rely on modern LLMs.

Figure 14 shows that, on tables with both numerical and text features, TARTE-FT is competitive with TabPFNv2 operating on non-pretrained representations. However, it is outperformed by knowledge-rich embeddings from Llama-3.1-8B, used as inputs for Ridge or TabPFNv2. Once again, for tabular learning with text, the largest gains come from knowledge-rich text representations, rather than architectural sophistication alone, highlighting the need for future table foundation models that leverage LLM-based text representations.

## D.4 OVERALL MODEL RANKING

Figure 15 presents a critical difference diagram comparing the mean ranks of all embedding methods when paired with a Ridge predictor. It also includes the performance of more advanced estimators on non-pretrained representations for context.

## D.5 PERFORMANCE ANALYSIS BY DATA SOURCE

Figure 16 illustrates the relative improvements of knowledge-rich representations over non-pretrained ones, broken down by data source. The benefits of external knowledge vary with dataset characteristics; tables from WikiDBs and CARTE, which are more knowledge-intensive, gain more from these representations than those from TextTabBench.

Figure 17 details the effect of LLM size on performance for each data source, confirming the scaling trend across different types of tables.

Figure 18 compares the performance of base LLMs to their counterparts refined on KGs. The benefits of refinement are most pronounced for the WikiDBs datasets, which are inherently more knowledge-centric as they are derived from a knowledge base.

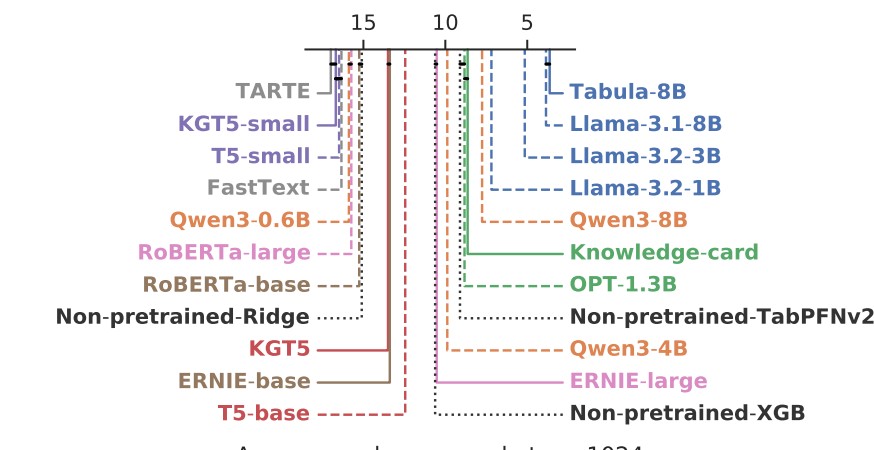

Average rank measured at $n = 1024$

Figure 15: Critical difference diagram across all data sources and methods.

## DECLARATION OF LLM USAGE

LLMs were used to polish the writing of some parts of this paper.

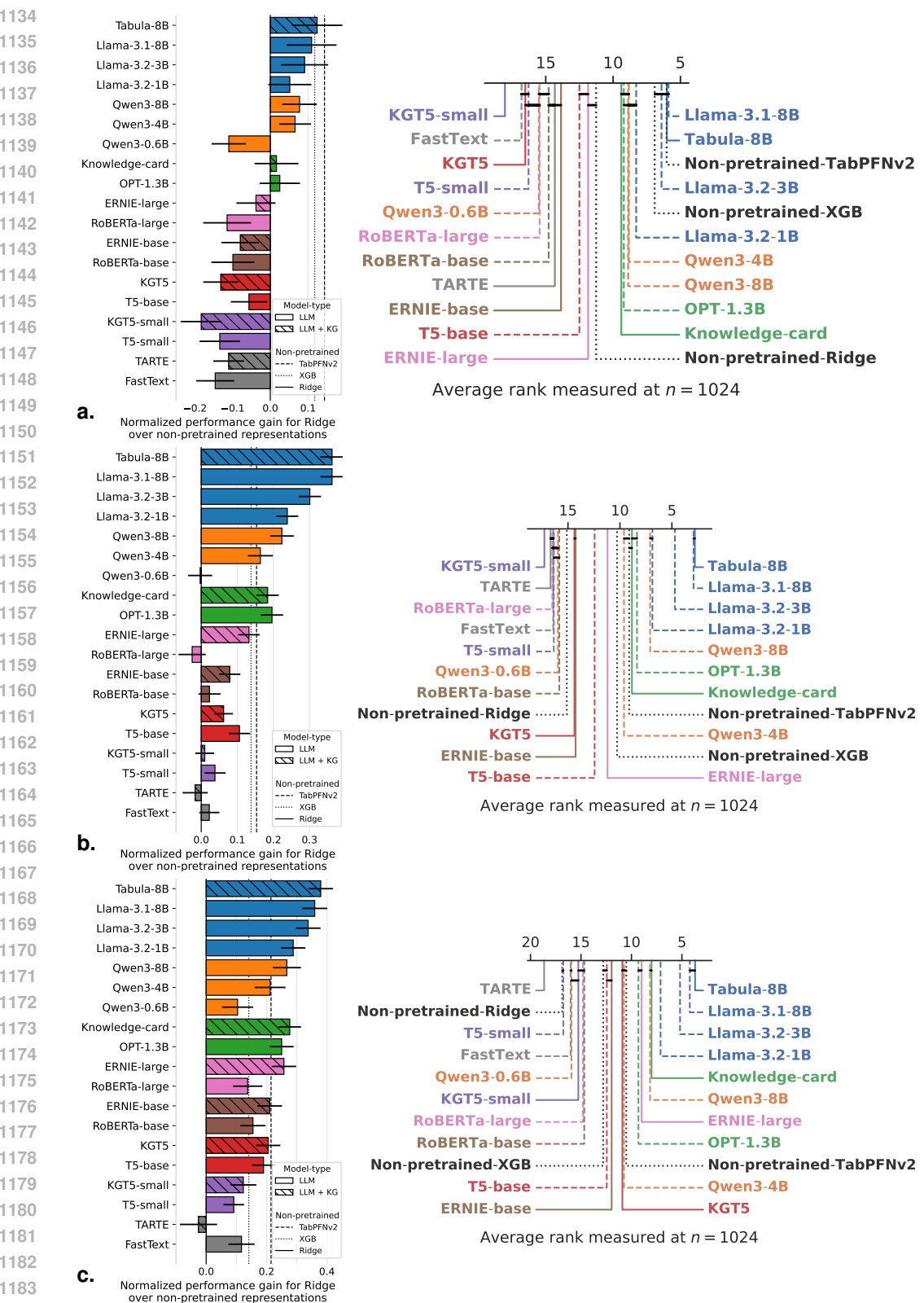

Figure 16: Relative improvements to non-pretrained string representations, when using a Ridge model as a tabular learner. For each source, larger models consistently yield better performances: **a.** TextTabBench **b.** CARTE **c.** WikiDBs.

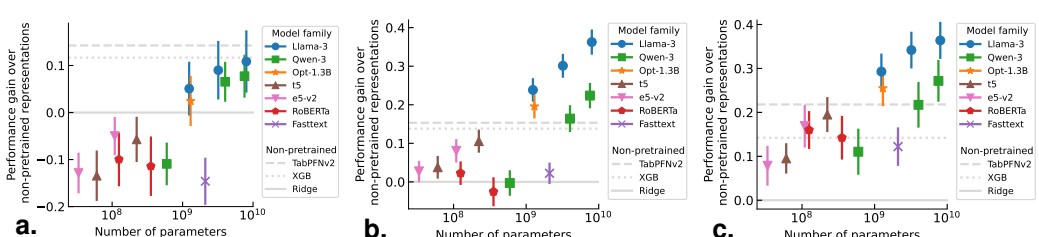

Figure 17: Effect of the size of representations from pure-LLM models for each source: **a.** Text-TabBench **b.** CARTE **c.** WikiDBs.

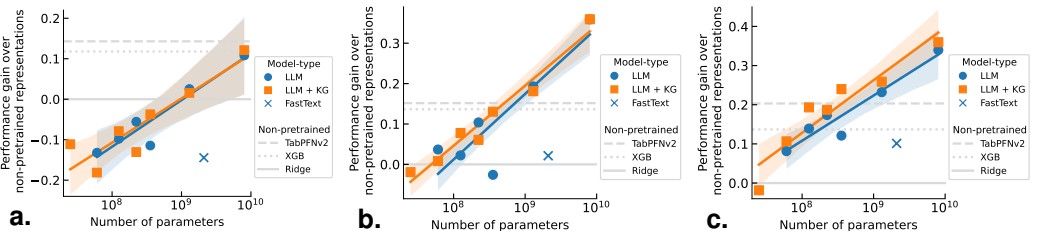

Figure 18: Comparison of LLM and their matched counterpart refined on knowledge bases for each source: **a.** TextTabBench **b.** CARTE **c.** WikiDBs.

Table 7: Details of the datasets used for automatic entity linking with BLINK, including the columns that were linked, the proportion of entries linked, and the total time taken for linking.

| Dataset | Columns linked (proportion linked) | Time |
|---|---|---|
| Airbnb | `neighbourhood` (85%), `property_type` (99%), `smart_location` (100%) | 38m |
| Customer Complaints | `Company` (93%), `State` (88%) | 8m |
| IT Salary | `Gender` (100%), `City` (100%), `Position` (72%), `Seniority level` (95%), `Your main technology/programming language` (97%), `Company type` (98%) | 21m |
| Mercari | `category_name` (60%), `brand_name` (68%) | 1h 8m |
| Osha Accidents | `Nature of Injury` (50%), `Part of Body` (81%) | 23m |
| Wine | `Grape` (99%), `Closure` (66%), `Country` (100%), `Type` (100%), `Region` (92%), `Appellation` (74%) | 23m |
| Babies R Us | `company_struct` (49%) | 14m |
| Bikedekho | `bike_name` (85%), `city_posted` (100%) | 27m |
| Bikewale | `bike_name` (91%), `city_posted` (100%) | 52m |
| Chocolate Bar Ratings | `Company_(Manufacturer)` (63%), `Company_Location` (95%), `Country_of_Bean_Origin` (100%) | 19m |
| Coffee Ratings | `roaster` (35%), `location` (94%), `origin` (87%) | 14m |
| Employee Salaries | `department_name` (81%), `division` (50%), `employee_position_title` (58%) | 1h 16m |
| Michelin | `Name` (59%), `Location` (96%), `Cuisine` (87%) | 56m |
| NBA Draft | `team` (62%), `player` (97%), `college` (74%) | 12m |
| Ramen Ratings | `Brand` (70%), `Variety` (67%), `Style` (91%), `Country` (100%) | 40m |
| Rotten Tomatoes | `Name` (91%), `Director` (90%), `Country` (98%), `Genre` (72%) | 1h 20m |
| Used Cars 24 | `Car_Brand` (77%), `Model` (70%) | 32m |
| UsedCars.com | `brand` (98%), `model` (88%) | 21m |
| Used Cars Saudi Arabia | `Region` (98%), `Model` (96%) | 29m |
| Artist Copyrights | `ArtistName` (50%), `ArtistCountryOfCitizenship` (79%) | 10m |
| Forward Players | `PLAYER_LABEL` (82%), `NATIONALITY` (98%) | 7m |
| Artworks Catalog | `MuseumLocation` (94%), `ArtistName` (78%), `ArtistCountry` (100%) | 9m |
| Geographers | `Full_Name` (55%), `Professional_Role` (100%), `Birth_Location` (69%), `Nationality` (85%) | 11m |
| Research Articles | `research_topic` (79%), `author_full_name` (39%), `primary_author` (30%) | 1h 5m |
| Sculptures | `Collection_Name` (74%), `Artist_Name` (78%) | 20m |
| Spring Locations | `SpringName` (43%), `AdministrativeEntity` (95%) | 31m |
| Geopolitical Regions | `region_name` (98%) | 2m |
| Kindergarten Locations | `City` (96%) | 7m |
| Sub Post Offices | `SUB_POST_OFFICE_NAME` (68%), `POSTAL_DIVISION` (70%) | 8m |
| State Schools | `SchoolName` (56%), `AdministrativeRegion` (93%) | 14m |
| Parish Churches | `ChurchName` (61%), `AdministrativeEntity` (95%) | 7m |
| Registered Ships | `ShipName` (57%), `ShipType` (100%), `Shipbuilder` (84%), `RegistryCountry` (100%), `HomePort` (95%) | 59m |
| Philosophers | `FullName` (69%), `BirthPlace` (91%), `Profession` (99%) | 34m |

Table 8: Overview of TextTabBench datasets used in our benchmark. Table statistics after preprocessing.

| Dataset | Task | # rows | # columns | | # classes | # linked | BLINK |
|---------|------|--------|-----------|------|-----------|----------|-------|
| | | | *cat.* | *num.* | | | |
| Diabetes | b-clf | 17,000 | 4 | 12 | 2 | - | - |
| Job Frauds | b-clf | 1,732 | 11 | 3 | 2 | - | - |
| Kickstarter | b-clf | 18,720 | 9 | 8 | 2 | - | - |
| Lending Club | b-clf | 11,254 | 12 | 15 | 2 | - | - |
| Osha Accidents | b-clf | 3,598 | 15 | 3 | 2 | - | ✓ |
| Customer Complaints | m-clf | 1,384 | 8 | 2 | 4 | - | ✓ |
| Spotify | m-clf | 10,000 | 3 | 15 | 10 | - | - |
| Airbnb | reg | 3,818 | 32 | 23 | - | - | ✓ |
| Beer | reg | 2,914 | 5 | 15 | - | - | - |
| California Houses | reg | 11,349 | 13 | 16 | - | - | - |
| Covid Trials | reg | 1,165 | 13 | 2 | - | - | - |
| Insurance Complaints | reg | 37,484 | 8 | 2 | - | - | - |
| IT Salary | reg | 1,253 | 16 | 2 | - | - | ✓ |
| Mercari | reg | 12,000 | 4 | 2 | - | - | ✓ |
| San Francisco Permits | reg | 183,794 | 12 | 16 | - | - | - |
| Stack Overflow | reg | 19,427 | 89 | 13 | - | - | - |
| Wine | reg | 1,281 | 12 | 3 | - | - | ✓ |

Table 9: Overview of CARTE datasets used in our benchmark. Table statistics after preprocessing.

| Dataset | Task | # rows | # columns | | # classes | # linked | BLINK |
|---|---|---|---|---|---|---|---|
| | | | *cat.* | *num.* | | | |
| Chocolate Bar Ratings | b-clf | 2,218 | 6 | 5 | 2 | - | ✓ |
| Coffee Ratings | b-clf | 1,670 | 8 | - | 2 | - | ✓ |
| Michelin | b-clf | 6,774 | 5 | 2 | 2 | - | ✓ |
| NBA Draft | b-clf | 1,550 | 4 | 5 | 2 | - | ✓ |
| Ramen Ratings | b-clf | 3,726 | 4 | - | 2 | - | ✓ |
| Roger Ebert | b-clf | 2,668 | 5 | 5 | 2 | - | - |
| Spotify | b-clf | 41,096 | 7 | 11 | 2 | - | - |
| US Accidents Severity | b-clf | 20,930 | 9 | 4 | 2 | - | - |
| Whisky | b-clf | 1,788 | 6 | - | 2 | - | - |
| Yelp | b-clf | 60,088 | 8 | 4 | 2 | - | - |
| Zomato | b-clf | 60,302 | 7 | 1 | 2 | - | - |
| Movies | reg | 7,224 | 7 | 9 | - | 7,095 | - |
| US Accidents Counts | reg | 22,623 | 6 | - | - | 14,697 | - |
| US Presidential | reg | 19,857 | 6 | - | - | 13,221 | - |
| Anime Planet | reg | 14,391 | 6 | 10 | - | - | - |
| Babies R Us | reg | 5,085 | 4 | - | - | - | ✓ |
| Beer Ratings | reg | 3,197 | 5 | 14 | - | - | - |
| Bikedekho | reg | 4,786 | 5 | 5 | - | - | ✓ |
| Bikewale | reg | 8,992 | 5 | 5 | - | - | ✓ |
| Buy Buy Baby | reg | 10,718 | 4 | - | - | - | - |
| Cardekho | reg | 37,813 | 13 | 6 | - | - | - |
| Clear Corpus | reg | 4,724 | 10 | 19 | - | - | - |
| Company Employees | reg | 10,941 | 7 | - | - | - | - |
| Employee Remuneration | reg | 35,396 | 2 | 5 | - | - | - |
| Employee Salaries | reg | 9,211 | 6 | 8 | - | - | ✓ |
| Fifa22 Players | reg | 18,085 | 9 | 18 | - | - | - |
| Filmtv Movies | reg | 41,205 | 6 | 6 | - | - | - |
| Journal JCR | reg | 9,615 | 4 | 5 | - | - | - |
| Journal SJR | reg | 27,931 | 9 | - | - | - | - |
| Japanese Anime | reg | 15,535 | 11 | 5 | - | - | - |
| K-Drama | reg | 1,239 | 8 | 4 | - | - | - |
| ML/DS Salaries | reg | 10,456 | 7 | 4 | - | - | - |
| Museums | reg | 11,467 | 14 | 2 | - | - | - |
| Mydramalist | reg | 3,400 | 10 | 3 | - | - | - |
| Prescription Drugs | reg | 1,714 | 5 | 5 | - | - | - |
| Rotten Tomatoes | reg | 7,158 | 10 | 6 | - | - | ✓ |
| Used Cars 24 | reg | 5,918 | 6 | 5 | - | - | ✓ |
| Used Cars Benz Italy | reg | 16,391 | 5 | 2 | - | - | - |
| UsedCars.com | reg | 4,009 | 8 | 5 | - | - | ✓ |
| Used Cars Pakistan | reg | 72,655 | 4 | 6 | - | - | - |
| Used Cars Saudi Arabia | reg | 5,507 | 7 | 6 | - | - | ✓ |
| Videogame Sales | reg | 16,410 | 4 | 4 | - | - | - |
| Wikiliq Beer | reg | 13,461 | 7 | 2 | - | - | - |
| Wikiliq Spirit | reg | 12,275 | 5 | 2 | - | - | - |
| Wina Poland | reg | 2,247 | 12 | 6 | - | - | - |
| Wine.com Prices | reg | 15,254 | 6 | 3 | - | - | - |
| Wine.com Ratings | reg | 4,095 | 6 | 3 | - | - | - |
| WineEnthusiasts Prices | reg | 120,975 | 8 | 1 | - | - | - |
| WineEnthusiasts Ratings | reg | 129,971 | 8 | 1 | - | - | - |
| WineVivino Price | reg | 13,834 | 5 | 2 | - | - | - |
| WineVivino Rating | reg | 13,834 | 6 | 2 | - | - | - |

Table 10: Overview of WikiDBs datasets used in our benchmark. Table statistics after preprocessing.

| Dataset | Task | # rows | # columns cat. | # columns num. | # classes | # linked | BLINK |
|---|---|---|---|---|---|---|---|
| CC Authors | b-clf | 16,224 | 7 | 1 | 2 | 1,302 | - |
| Defenders | m-clf | 18,610 | 10 | - | 10 | 8,700 | - |
| Philosophers | m-clf | 4,230 | 8 | - | 10 | 1,656 | ✓ |
| US Music Albums | m-clf | 3,270 | 10 | 1 | 10 | 2,180 | - |
| Artist Copyrights | m-clf | 2,000 | 9 | 1 | 10 | - | ✓ |
| Artworks Catalog | m-clf | 1,210 | 8 | 2 | 10 | - | ✓ |
| Forward Players | m-clf | 1,400 | 10 | - | 10 | - | ✓ |
| Geographers | m-clf | 1,130 | 9 | - | 10 | - | ✓ |
| Historic Buildings | m-clf | 27,980 | 6 | 3 | 10 | - | - |
| Island | m-clf | 19,650 | 3 | 2 | 10 | - | - |
| Kindergarten Locations | m-clf | 2,790 | 6 | - | 3 | - | ✓ |
| Magic Narratives | m-clf | 1,062 | 4 | - | 9 | - | - |
| Museums | m-clf | 9,550 | 4 | 2 | 10 | - | - |
| Noble Individuals | m-clf | 1,400 | 9 | - | 10 | - | - |
| Notable Trees | m-clf | 1,408 | 4 | 2 | 8 | - | - |
| Parish Churches | m-clf | 1,350 | 4 | 2 | 10 | - | ✓ |
| Sculptures | m-clf | 3,720 | 6 | - | 10 | - | ✓ |
| Spring Locations | m-clf | 5,930 | 2 | 2 | 10 | - | ✓ |
| State Schools | m-clf | 2,800 | 3 | 2 | 10 | - | ✓ |
| Scientific Articles | m-clf | 2,760 | 13 | 1 | 10 | - | - |
| Sub Post Offices | m-clf | 1,530 | 3 | 1 | 10 | - | ✓ |
| Transport Stations | m-clf | 4,640 | 8 | 2 | 10 | - | - |
| Business Locations | reg | 16,821 | 4 | 4 | - | 16,438 | - |
| Dissolved Municipalities | reg | 13,462 | 6 | 2 | - | 1,656 | - |
| Geopolitical Regions | reg | 1,114 | 6 | 3 | - | 1,066 | ✓ |
| Historical Figures | reg | 11,260 | 11 | - | - | 2,134 | - |
| Municipal District Capitals | reg | 1,658 | 5 | 3 | - | 1,267 | - |
| Poets | reg | 60,240 | 10 | - | - | 21,564 | - |
| Territorial Entities | reg | 36,717 | 7 | 4 | - | 34,189 | - |
| WWI Personnel | reg | 30,675 | 11 | - | - | 16,227 | - |
| Artworks Inventory | reg | 10,635 | 5 | 1 | - | - | - |
| Drawings Catalog | reg | 63,130 | 8 | 1 | - | - | - |
| Eclipsing Binary Stars | reg | 297,934 | 6 | 2 | - | - | - |
| Registered Ships | reg | 4,644 | 6 | 3 | - | - | ✓ |
| Research Articles | reg | 6,962 | 6 | 2 | - | - | ✓ |
| Research Article Citations | reg | 4,115 | 9 | - | - | - | - |
| Ukrainian Villages | reg | 21,355 | 3 | 3 | - | - | - |

Table 11: A snippet from the `Customer Complaints` dataset, where LLMs perform well. The task is to predict the "Company response to consumer" (shortened to "Company response" here for space reasons). Some columns were removed to fit the table in the paper.

| Issue | Product | Company | Submitted via | State | Company response |
|---|---|---|---|---|---|
| Incorrect information on credit report | Credit reporting | Experian Information Solutions Inc. | Web | CO | 0 |
| Written notification about debt | Debt collection | Associated Credit Services, Inc. | Web | NY | 0 |
| Struggling to pay mortgage | Mortgage | RoundPoint Mortgage Servicing Corporation | Web | NY | 0 |

Table 12: A snippet from the `Fifa22 Players` dataset, where LLMs refined on KGs perform well. The task is to predict the player's wage. Some columns were removed to fit the table in the paper.

| name | club_name | player_positions | nationality_name | work_rate | body_type | wage_eur |
|------|-----------|------------------|------------------|-----------|-----------|----------|
| L. Cass | Port Vale | CB, RB | England | High/Medium | Lean (185+) | 3.602060 |
| Judson | San Jose Earthquakes | CDM | Brazil | Medium/High | Normal (170-) | 3.778151 |
| E. Gyasi | Spezia | RW, LW, ST | Ghana | High/Low | Lean (170-185) | 3.845098 |
| Z. Kvržić | Yukatel Kayserispor | RB, CAM, RM | Bosnia and Herzegovina | Medium/Mediu | Lean (170-185) | 3.477121 |

Table 13: A snippet from the `Registered Ships` dataset, where column-context brings value. The task is to predict the gross tonnage.

| RegistryCountry | HomePort | ShipName | Shipbuilder | ShipType | GrossTonnage |
|-----------------|----------|----------|-------------|----------|--------------|
| Liberia | Monrovia | A Whale | Hyundai Heavy Industries | ore-bulk-oil carrier | 5.230242 |
| Liberia | Nassau | Adventure of the Seas | Kvaerner Masa-Yards | cruise ship | 5.137595 |
| Liberia | Monrovia | IMO 9225615 | Hanwha Ocean | container ship | 4.878464 |
| Norway | NaN | Serenissima | Trondhjems mekaniske Værksted | motor ship | 3.414639 |
| Liberia | NaN | Otto Hahn | Howaldtswerke-Deutsche Werft | ship | 4.211948 |

Table 14: A snippet from the `Company Employees` dataset, where column-context hurts performance. The task is to predict the current employee estimate.

| industry | locality | name | domain | current_employee_estimate |
|----------|----------|------|--------|---------------------------|
| information technology and services | new york, new york, united states | ibm | ibm.com | 5.437825 |
| information technology and services | bombay, maharashtra, india | tata consultancy services | tcs.com | 5.280512 |
| information technology and services | dublin, dublin, ireland | accenture | accenture.com | 5.280326 |
| accounting | london, greater london, united kingdom | ey | ey.com | 5.199654 |

Table 15: Average runtimes (in seconds) for embedding extraction and Ridge fitting, for varying train set sizes.

| | Train size | | |
|---|---|---|---|
| | 64 | 256 | 1,024 |
| TabuLa-8B | $124 \pm_{141}$ | $145 \pm_{166}$ | $216 \pm_{258}$ |
| Llama-3.1-8B | $119 \pm_{133}$ | $140 \pm_{157}$ | $209 \pm_{247}$ |
| Llama-3.2-3B | $43 \pm_{51}$ | $51 \pm_{60}$ | $76 \pm_{93}$ |
| Llama-3.2-1B | $18 \pm_{20}$ | $21 \pm_{24}$ | $32 \pm_{37}$ |
| Qwen3-8B | $120 \pm_{144}$ | $140 \pm_{169}$ | $210 \pm_{262}$ |
| Qwen3-4B | $65 \pm_{82}$ | $76 \pm_{96}$ | $114 \pm_{150}$ |
| Qwen3-0.6B | $12 \pm_{13}$ | $14 \pm_{15}$ | $21 \pm_{22}$ |
| Knowledge-card | $25 \pm_{29}$ | $30 \pm_{34}$ | $45 \pm_{53}$ |
| OPT-1.3B | $23 \pm_{29}$ | $27 \pm_{34}$ | $40 \pm_{53}$ |
| ERNIE-large | $8 \pm_6$ | $10 \pm_7$ | $15 \pm_9$ |
| RoBERTa-large | $8 \pm_6$ | $10 \pm_7$ | $14 \pm_9$ |
| ERNIE-base | $5 \pm_4$ | $6 \pm_4$ | $8 \pm_6$ |
| RoBERTa-base | $4 \pm_3$ | $5 \pm_3$ | $7 \pm_4$ |
| KGT5 | $5 \pm_3$ | $6 \pm_4$ | $8 \pm_5$ |
| T5-base | $5 \pm_6$ | $7 \pm_7$ | $9 \pm_{11}$ |
| KGT5-small | $3 \pm_3$ | $4 \pm_3$ | $6 \pm_4$ |
| T5-small | $4 \pm_3$ | $4 \pm_3$ | $6 \pm_4$ |
| TARTE | $4 \pm_4$ | $5 \pm_5$ | $8 \pm_6$ |
| FastText | $2 \pm_4$ | $3 \pm_4$ | $4 \pm_6$ |
| Non-pretrained | $0.5 \pm_{0.7}$ | $1 \pm_1$ | $2 \pm_2$ |

Table 16: Raw results on TextTabBench datasets. Mean and standard error over 17 datasets and 10 random seeds, for train size $n_{\text{train}} = 1,024$.

| | Regression (R2) | | | Classification (ROC AUC) | | |
|---|---|---|---|---|---|---|
| | Ridge | TabPFNv2 (PCA $d = 500$) | XGBoost (PCA $d = 300$) | Ridge | TabPFNv2 (PCA $d = 500$) | XGBoost (PCA $d = 300$) |
| TabuLa-8B | $0.409 \pm .024$ | $0.404 \pm .023$ | $0.358 \pm .021$ | $\mathbf{0.813} \pm \mathbf{.017}$ | $0.789 \pm .018$ | $0.792 \pm .018$ |
| LLaMA-3.1-8B | $0.398 \pm .024$ | $0.392 \pm .023$ | $0.349 \pm .020$ | $0.807 \pm .017$ | $0.782 \pm .018$ | $0.782 \pm .018$ |
| LLaMA-3.2-3B | $0.396 \pm .023$ | $0.393 \pm .022$ | $0.350 \pm .020$ | $0.802 \pm .017$ | $0.783 \pm .018$ | $0.783 \pm .018$ |
| LLaMA-3.2-1B | $0.366 \pm .021$ | $0.359 \pm .020$ | $0.321 \pm .018$ | $0.803 \pm .016$ | $0.779 \pm .018$ | $0.786 \pm .017$ |
| Qwen3-8B | $\mathbf{0.411} \pm \mathbf{.018}$ | $0.383 \pm .019$ | $0.345 \pm .016$ | $0.800 \pm .017$ | $0.775 \pm .018$ | $0.780 \pm .018$ |
| Qwen3-4B | $0.397 \pm .017$ | $0.367 \pm .018$ | $0.330 \pm .017$ | $0.797 \pm .016$ | $0.774 \pm .018$ | $0.777 \pm .018$ |
| Qwen3-0.6B | $0.303 \pm .015$ | $0.280 \pm .015$ | $0.256 \pm .013$ | $0.776 \pm .015$ | $0.750 \pm .018$ | $0.760 \pm .017$ |
| Knowledge-card | $0.333 \pm .019$ | $0.332 \pm .019$ | $0.291 \pm .016$ | $0.804 \pm .016$ | $0.780 \pm .018$ | $0.784 \pm .018$ |
| OPT-1.3B | $0.350 \pm .019$ | $0.347 \pm .017$ | $0.304 \pm .016$ | $0.798 \pm .017$ | $0.777 \pm .018$ | $0.781 \pm .018$ |
| ERNIE-large | $0.323 \pm .017$ | $0.312 \pm .015$ | $0.270 \pm .013$ | $0.790 \pm .016$ | $0.766 \pm .019$ | $0.770 \pm .018$ |
| RoBERTa-large | $0.265 \pm .016$ | $0.300 \pm .018$ | $0.262 \pm .016$ | $0.789 \pm .016$ | $0.770 \pm .018$ | $0.774 \pm .018$ |
| ERNIE-base | $0.307 \pm .017$ | $0.299 \pm .014$ | $0.264 \pm .013$ | $0.785 \pm .016$ | $0.762 \pm .018$ | $0.766 \pm .017$ |
| RoBERTa-base | $0.279 \pm .017$ | $0.279 \pm .017$ | $0.238 \pm .015$ | $0.783 \pm .016$ | $0.762 \pm .018$ | $0.766 \pm .018$ |
| KGT5 | $0.270 \pm .017$ | $0.258 \pm .017$ | $0.228 \pm .014$ | $0.773 \pm .016$ | $0.759 \pm .018$ | $0.755 \pm .018$ |
| T5 | $0.312 \pm .017$ | $0.315 \pm .016$ | $0.273 \pm .014$ | $0.787 \pm .015$ | $0.765 \pm .018$ | $0.767 \pm .018$ |
| E5-v2 | $0.314 \pm .015$ | $0.295 \pm .016$ | $0.271 \pm .014$ | $0.789 \pm .015$ | $0.762 \pm .018$ | $0.769 \pm .018$ |
| KGT5-small | $0.242 \pm .015$ | $0.223 \pm .015$ | $0.209 \pm .013$ | $0.766 \pm .015$ | $0.751 \pm .018$ | $0.754 \pm .017$ |
| T5-small | $0.260 \pm .017$ | $0.265 \pm .016$ | $0.227 \pm .014$ | $0.778 \pm .015$ | $0.756 \pm .017$ | $0.761 \pm .017$ |
| E5-small-v2 | $0.278 \pm .015$ | $0.281 \pm .016$ | $0.244 \pm .014$ | $0.776 \pm .016$ | $0.770 \pm .018$ | $0.754 \pm .019$ |
| TARTE | $0.320 \pm .017$ | $0.314 \pm .018$ | $0.278 \pm .016$ | $0.778 \pm .014$ | $0.758 \pm .017$ | $0.758 \pm .016$ |
| FastText | $0.278 \pm .017$ | $0.320 \pm .016$ | $0.278 \pm .015$ | $0.770 \pm .016$ | $0.775 \pm .017$ | $0.766 \pm .018$ |
| Non-pretrained | $0.379 \pm .018$ | $\mathbf{0.452} \pm \mathbf{.020}$ | $\mathbf{0.440} \pm \mathbf{.020}$ | $0.781 \pm .015$ | $\mathbf{0.804} \pm \mathbf{.016}$ | $\mathbf{0.800} \pm \mathbf{.015}$ |

Table 17: Raw results on CARTE datasets. Mean and standard error over 51 datasets and 10 random seeds, for train size $n_{\text{train}} = 1,024$.

| | Regression (R2) | | | Classification (ROC AUC) | | |
|---|---|---|---|---|---|---|
| | Ridge | TabPFNv2 (PCA $d = 500$) | XGBoost (PCA $d = 300$) | Ridge | TabPFNv2 (PCA $d = 500$) | XGBoost (PCA $d = 300$) |
| TabuLa-8B | $0.608 \pm .009$ | $0.587 \pm .009$ | $0.550 \pm .009$ | $\mathbf{0.824} \pm \mathbf{.008}$ | $\mathbf{0.797} \pm \mathbf{.009}$ | $0.793 \pm .008$ |
| LLaMA-3.1-8B | $\mathbf{0.609} \pm \mathbf{.008}$ | $\mathbf{0.587} \pm \mathbf{.009}$ | $\mathbf{0.550} \pm \mathbf{.008}$ | $0.823 \pm .008$ | $0.796 \pm .009$ | $\mathbf{0.793} \pm \mathbf{.008}$ |
| LLaMA-3.2-3B | $0.582 \pm .009$ | $0.564 \pm .010$ | $0.529 \pm .009$ | $0.812 \pm .008$ | $0.783 \pm .009$ | $0.780 \pm .008$ |
| LLaMA-3.2-1B | $0.552 \pm .009$ | $0.541 \pm .010$ | $0.505 \pm .009$ | $0.806 \pm .008$ | $0.777 \pm .009$ | $0.776 \pm .008$ |
| Qwen3-8B | $0.546 \pm .009$ | $0.513 \pm .010$ | $0.480 \pm .009$ | $0.799 \pm .008$ | $0.767 \pm .009$ | $0.769 \pm .008$ |
| Qwen3-4B | $0.525 \pm .009$ | $0.503 \pm .010$ | $0.472 \pm .010$ | $0.782 \pm .007$ | $0.748 \pm .008$ | $0.754 \pm .008$ |
| Qwen3-0.6B | $0.460 \pm .010$ | $0.432 \pm .011$ | $0.413 \pm .010$ | $0.743 \pm .008$ | $0.711 \pm .008$ | $0.717 \pm .008$ |
| Knowledge-card | $0.531 \pm .010$ | $0.512 \pm .010$ | $0.482 \pm .009$ | $0.793 \pm .007$ | $0.761 \pm .008$ | $0.762 \pm .008$ |
| OPT-1.3B | $0.533 \pm .009$ | $0.514 \pm .010$ | $0.482 \pm .009$ | $0.800 \pm .008$ | $0.768 \pm .009$ | $0.773 \pm .008$ |
| ERNIE-large | $0.509 \pm .009$ | $0.503 \pm .010$ | $0.468 \pm .009$ | $0.782 \pm .008$ | $0.744 \pm .009$ | $0.750 \pm .008$ |
| RoBERTa-large | $0.429 \pm .008$ | $0.497 \pm .010$ | $0.463 \pm .009$ | $0.774 \pm .008$ | $0.745 \pm .009$ | $0.752 \pm .008$ |
| ERNIE-base | $0.487 \pm .009$ | $0.484 \pm .011$ | $0.453 \pm .010$ | $0.772 \pm .008$ | $0.734 \pm .009$ | $0.740 \pm .009$ |
| RoBERTa-base | $0.458 \pm .009$ | $0.479 \pm .011$ | $0.447 \pm .010$ | $0.771 \pm .008$ | $0.740 \pm .009$ | $0.746 \pm .008$ |
| KGT5 | $0.480 \pm .010$ | $0.469 \pm .011$ | $0.438 \pm .010$ | $0.761 \pm .008$ | $0.725 \pm .009$ | $0.732 \pm .008$ |
| T5 | $0.503 \pm .010$ | $0.486 \pm .011$ | $0.454 \pm .010$ | $0.771 \pm .009$ | $0.733 \pm .010$ | $0.741 \pm .009$ |
| E5-v2 | $0.488 \pm .010$ | $0.476 \pm .011$ | $0.448 \pm .010$ | $0.774 \pm .007$ | $0.737 \pm .008$ | $0.752 \pm .008$ |
| KGT5-small | $0.458 \pm .010$ | $0.450 \pm .011$ | $0.419 \pm .010$ | $0.752 \pm .008$ | $0.717 \pm .009$ | $0.724 \pm .008$ |
| T5-small | $0.476 \pm .010$ | $0.461 \pm .012$ | $0.434 \pm .010$ | $0.759 \pm .009$ | $0.720 \pm .009$ | $0.729 \pm .008$ |
| E5-small-v2 | $0.467 \pm .010$ | $0.471 \pm .011$ | $0.433 \pm .010$ | $0.761 \pm .008$ | $0.751 \pm .008$ | $0.737 \pm .008$ |
| TARTE | $0.451 \pm .010$ | $0.449 \pm .012$ | $0.417 \pm .010$ | $0.746 \pm .008$ | $0.714 \pm .008$ | $0.726 \pm .008$ |
| FastText | $0.464 \pm .010$ | $0.496 \pm .011$ | $0.471 \pm .010$ | $0.763 \pm .008$ | $0.754 \pm .008$ | $0.753 \pm .008$ |
| Non-pretrained | $0.430 \pm .011$ | $0.526 \pm .011$ | $0.519 \pm .010$ | $0.766 \pm .008$ | $0.773 \pm .008$ | $0.768 \pm .008$ |

Table 18: Raw results on WikiDBs datasets. Mean and standard error over 37 datasets and 10 random seeds, for train size $n_{\text{train}} = 1,024$.

| | Regression (R2) | | | Classification (ROC AUC) | | |
|---|---|---|---|---|---|---|
| | Ridge | TabPFNv2 (PCA $d = 500$) | XGBoost (PCA $d = 300$) | Ridge | TabPFNv2 (PCA $d = 500$) | XGBoost (PCA $d = 300$) |
| TabuLa-8B | **0.552** ± **.018** | **0.546** ± **.019** | **0.492** ± **.017** | **0.954** ± **.005** | **0.951** ± **.006** | **0.947** ± **.006** |
| LLaMA-3.1-8B | 0.542 ± .017 | 0.530 ± .018 | 0.480 ± .017 | 0.950 ± .006 | 0.946 ± .006 | 0.942 ± .006 |
| LLaMA-3.2-3B | 0.516 ± .018 | 0.504 ± .020 | 0.459 ± .018 | 0.951 ± .005 | 0.948 ± .006 | 0.944 ± .006 |
| LLaMA-3.2-1B | 0.486 ± .016 | 0.479 ± .018 | 0.431 ± .016 | 0.945 ± .006 | 0.942 ± .006 | 0.938 ± .006 |
| Qwen3-8B | 0.468 ± .018 | 0.456 ± .020 | 0.404 ± .017 | 0.946 ± .006 | 0.943 ± .007 | 0.937 ± .007 |
| Qwen3-4B | 0.439 ± .018 | 0.432 ± .019 | 0.386 ± .017 | 0.942 ± .006 | 0.940 ± .007 | 0.936 ± .007 |
| Qwen3-0.6B | 0.376 ± .014 | 0.356 ± .016 | 0.320 ± .014 | 0.927 ± .007 | 0.926 ± .007 | 0.920 ± .007 |
| Knowledge-card | 0.476 ± .015 | 0.470 ± .016 | 0.428 ± .014 | 0.945 ± .006 | 0.944 ± .006 | 0.940 ± .006 |
| OPT-1.3B | 0.465 ± .016 | 0.456 ± .017 | 0.409 ± .015 | 0.943 ± .006 | 0.940 ± .006 | 0.936 ± .006 |
| ERNIE-large | 0.475 ± .016 | 0.475 ± .017 | 0.425 ± .015 | 0.942 ± .006 | 0.940 ± .006 | 0.935 ± .006 |
| RoBERTa-large | 0.381 ± .014 | 0.442 ± .016 | 0.396 ± .014 | 0.937 ± .006 | 0.939 ± .007 | 0.934 ± .007 |
| ERNIE-base | 0.449 ± .016 | 0.455 ± .017 | 0.406 ± .015 | 0.937 ± .006 | 0.936 ± .006 | 0.931 ± .007 |
| RoBERTa-base | 0.403 ± .014 | 0.418 ± .016 | 0.374 ± .014 | 0.935 ± .006 | 0.935 ± .007 | 0.930 ± .007 |
| KGT5 | 0.423 ± .014 | 0.426 ± .016 | 0.383 ± .014 | 0.937 ± .006 | 0.935 ± .007 | 0.929 ± .007 |
| T5 | 0.426 ± .015 | 0.423 ± .016 | 0.379 ± .014 | 0.937 ± .006 | 0.934 ± .007 | 0.929 ± .007 |
| E5-v2 | 0.416 ± .016 | 0.406 ± .018 | 0.368 ± .015 | 0.933 ± .006 | 0.931 ± .007 | 0.927 ± .007 |
| KGT5-small | 0.380 ± .013 | 0.389 ± .015 | 0.346 ± .013 | 0.929 ± .007 | 0.928 ± .007 | 0.922 ± .007 |
| T5-small | 0.383 ± .014 | 0.386 ± .015 | 0.342 ± .013 | 0.930 ± .006 | 0.926 ± .007 | 0.920 ± .007 |
| E5-small-v2 | 0.374 ± .013 | 0.381 ± .015 | 0.328 ± .013 | 0.926 ± .007 | 0.929 ± .007 | 0.921 ± .007 |
| TARTE | 0.356 ± .013 | 0.362 ± .015 | 0.319 ± .013 | 0.927 ± .007 | 0.927 ± .007 | 0.920 ± .007 |
| FastText | 0.383 ± .014 | 0.416 ± .015 | 0.385 ± .014 | 0.929 ± .006 | 0.932 ± .007 | 0.926 ± .007 |
| Non-pretrained | 0.364 ± .014 | 0.460 ± .016 | 0.435 ± .015 | 0.924 ± .007 | 0.941 ± .006 | 0.934 ± .006 |

