# OpenReview forum: "Tabular Learning with Background Information: LLMs, Knowledge Graphs, or Both?"
_ICLR.cc/2026/Conference — Submitted to ICLR 2026_

### Official Review · Reviewer_NUSu · 2025-10-21

**Soundness:** 2
**Presentation:** 3
**Contribution:** 2
**Rating:** 2
**Confidence:** 4

**Summary:**

This paper presents a large-scale empirical study to determine the best source of background knowledge for tabular learning tasks, specifically focusing on textual data within tables. The authors compare representations derived from pure Large Language Models (LLMs), structured Knowledge Graphs (KGs), and hybrid models that refine LLMs on KGs. Using a newly assembled benchmark of 105 tabular datasets containing text, the study finds that knowledge-rich representations significantly boost predictive performance, even allowing simple linear models to outperform strong tabular baselines like XGBoost. A key finding is that larger LLMs provide greater gains. Furthermore, on a controlled subset of 15 tables with pre-solved entity linking, pure KG embeddings perform on par with LLMs of similar size. This leads the authors to conclude that the primary advantage of LLMs is not superior knowledge storage but their innate ability to solve the "symbol grounding" or entity linking problem. The paper advocates for hybrid LLM+KG approaches as the most promising future direction.

**Strengths:**

1. This paper addresses the important and practical problem of leveraging external knowledge from textual features in tabular data . The assembly of a new, large-scale benchmark of 105 datasets is a significant contribution to this area of research.
2. The study provides a large-scale, systematic comparison across a wide spectrum of models. This includes multiple LLM families (e.g., Llama-3, Qwen3, ROBERTa, T5) and sizes , various hybrid LLM+KG models (e.g., ERNIE, KGT5, Knowledge Card) , classic pure KG embedding methods, and several distinct downstream tabular learners (Ridge, XGBoost, TabPFNv2).
3. The paper clearly articulates the conceptual gap between traditional tabular learning and knowledge-rich modeling. This framing highlights why background knowledge matters for textual tables and situates the problem within the broader challenge of symbol grounding and structured reasoning, all presented with clear and precise exposition.

**Weaknesses:**

1. My most significant concern is the paper's narrow definition of its own problem domain. In Section 3.1, the authors state, "we remove all numerical columns to focus our study on text-based knowledge". This single methodological choice fundamentally changes the problem from heterogeneous tabular learning (the domain of XGBoost, TabPFNv2, and real-world tables) to a short-text prediction problem. This makes the central claim (Finding 1) that their method "outperforms strong tabular baselines" deeply problematic. The SOTA baselines (XGB, TabPFN) are being benchmarked in an artificial setting they were not designed for (text-only). A fair comparison would require a heterogeneous setup.
2. The SOTA comparison in the paper is incomplete. The authors claim to “markedly outperform strong tabular baselines,” yet their evaluation is limited to models such as XGBoost and TabPFNv2. More critically, the study omits comparisons with genuine state-of-the-art methods for deep tabular learning—such as **FT-Transformer**, **SAINT, CARTE**, or competitive **MLP-based baselines**—which are explicitly designed to handle the kind of heterogeneous tabular data that this paper excludes. Additionally, **LLM-based approaches** like **TabLLM** are not considered, further weakening the strength of the claimed performance advantages.
3. The paper's conclusions are drawn from an experimental setup heavily skewed towards small-data regimes. The methodology explicitly states, "To simulate small-data scenarios... we sample training sets of varying sizes, $n_{train}\in\{64, 256, 1024\}$"14. All major results and ranking diagrams (e.g., Fig. 7, 18) are reported at $n=1024$. While the rationale (external knowledge is most critical here) is valid, this is a "cherry-picked battlefield" that maximizes the utility of external knowledge and disadvantages GBDTs, which are known to excel as $n_{train}$ grows. The paper provides no evidence that these conclusions generalize to larger, more realistic training sets (e.g., $n_{train} > 10,000$).
4. The novelty of the second finding ("Refining LLMs on KGs is a promising combination") is limited. As the paper's own related work section details, this is a well-established research direction. The paper's contribution here is to validate this on their benchmark, but the finding that this refinement only "boosts models slightly"  makes this contribution feel incremental rather than a significant breakthrough.

**Questions:**

1. To make the comparison to SOTA tabular learners fair and the conclusions relevant to *tabular learning*, the authors should include experiments on the original, heterogeneous data. The most direct approach would be to **concatenate** the numerical features (which were removed) with the new text-based row embeddings. How do the results change in this (more realistic) setting?
2. Can the authors provide results on at least a medium-sized training set (e.g., $n_{train}=10240$) to demonstrate the generalizability of their findings?
3. The authors are encouraged to extend their SOTA comparison by including modern deep tabular and LLM-based baselines—such as FT-Transformer, SAINT, DANet, TabNet, or TabLLM. Including these would greatly strengthen the validity of the paper’s claims and situate the proposed method more clearly within the current landscape of tabular learning research.

---

> ### Author Response · Authors · 2025-11-27
> **Response to reviewer NUSu**
>
> We thank the reviewer for thoughtful feedback and constructive suggestions, highlighting the significance of our contribution, the breadth of our benchmark and model comparisons, and the clarity of our paper.
>
> Below, we respond to the reviewer’s questions.
>
> **Q1: Re-introducing numerical features**
>
> Our initial choice of excluding the numerical features was motivated by the question we wanted to answer: what is the best way to encode text features in tabular data? Removing numbers enable to isolate contributions from the textual features only.
>
> Nevertheless, we agree that **reintroducing the numerical features is a very interesting extension**, to study how these features interact with the textual features. Therefore, we conducted an additional experiment, where we re-included the numerical features by concatenating them to the text embeddings. Results reported (Sec. 5.2, Fig. 9, p. 10) show that:
> - The **relative ranking of text encoders remains unchanged.**
> - Combining numerical and textual features (num+str) markedly outperforms either modality alone, demonstrating that their **contributions are complementary rather than redundant.**
> - **Text-only features outperform numbers-only**, highlighting the primary importance of textual features.
> - Knowledge-rich representations provide huge benefits for Ridge, but little to a table foundation model like TabPFNv2, highlighting its shortcomings in leveraging fully high-dimensional knowledge-rich representations (this limitation is also supported by our new analysis of PCA in Sec. 5.1).
>
> TabPFNv2 is notably better than Ridge on numerical features. However our study shows that on embeddings of text features, it is the converse. When both numerical and text features, both approaches can be competitive, depending on the relative predictive importance of texts or numbers. However, our study shows that TabPFNv2 does not make the most of text features, opening an important research question for improving future table foundation models.
> We thank the reviewer for this suggestion, which allowed us to better articulate the implications of our study. We have updated our conclusion accordingly (updates in blue).
>
> **Q2: Evaluating on larger train sizes**
>
> We thank the reviewer for raising this important point. We initially limited our study to 1,024 samples since several of our (real-world) datasets do not go beyond this size. However, to assess whether our conclusions hold beyond the small-data regime, we ran **additional experiments on a subset of 49 datasets containing more than 10,000 rows**, and plotted learning curves covering the range 64 to 10,000 samples. The results are now included in Appendix B.1 (Fig. 11, p. 18).
>
> We find that while all models benefit from more data, the relative ordering between LLM, KG-refined models, and non-pretrained encoders remains consistent. In particular, knowledge-rich representations (LLM or KG-based) continue to outperform purely surface-form encodings even at 10k samples. These findings demonstrate that the advantages of LLM and KG embeddings are not confined to small-data regimes but persist across realistic training sizes, reinforcing the generality of our main conclusions.
>
> **Q3: Adding more baselines**
>
> We agree that more advanced learners could be studied. We choose TabPFNv2 as it is a leading model (according to TabArena). TabLLM uses the LLM in context, thus it cannot be scaled to the datasets that we investigate.
> Our computing resource did not allow us to run additional baselines on top of the other experiments we added, however we are happy to add some for the final version.

---

> > ### Author Response · Authors · 2025-12-02
> > **Adding more tabular baselines**
> >
> > We agree that including modern end-to-end tabular baselines, capable of jointly modeling strings and numbers, is interesting, especially now that our revised experiments reintroduce numerical features.
> >
> > Following the reviewer’s recommendation, **we added TARTE-FT** [1], the successor to CARTE and **a recent state-of-the-art model for heterogeneous tables**. We use the best version, a model fine-tuned on each specific downstream task (TARTE-FT), for enhanced performance.
> >
> > We now include TARTE-FT results in Appendix D.3 (p. 20, Fig. 14), showing that, on tables with both numerical and text features:
> > - TARTE-FT is competitive with TabPFNv2 operating on non-pretrained representations;
> > - however, it remains below knowledge-rich representations, such as Llama-3.1-8B embeddings used with Ridge or TabPFNv2.
> >
> > These results further reinforce our conclusion: for tabular learning with text, **the largest gains come from knowledge-rich text representations, rather than architectural sophistication alone**, highlighting the need for future table foundation models that leverage LLM-based text representations.
> >
> > Other baselines like FT-Transformer (2021), SAINT (2021), DANet (2022), or TabNet (2021), have been influential but are now several years old and no longer regarded as state-of-the-art in tabular learning (according to recent benchmarks such as TabArena).
> >
> > [1] Myung Jun Kim, Félix Lefebvre, Gaëtan Brison, Alexandre Perez-Lebel, and Gaël Varoquaux. Table foundation models: on knowledge pre-training for tabular learning. TMLR, 2025.

---

### Official Review · Reviewer_QSCG · 2025-10-29

**Soundness:** 3
**Presentation:** 2
**Contribution:** 2
**Rating:** 6
**Confidence:** 3

**Summary:**

This paper presents a large-scale empirical study on leveraging external background knowledge—specifically from large language models (LLMs) and knowledge graphs (KGs)—to improve tabular learning on datasets containing textual entity mentions. The authors assemble a benchmark of 105 real-world and semi-synthetic tabular datasets, systematically evaluate a wide range of representation methods (including pure LLMs, pure KG embeddings, and LLMs refined on KGs), and analyze performance under controlled settings—most notably using a subset of 15 tables with pre-linked Wikidata entities. Key findings include: (1) knowledge-rich representations significantly outperform traditional text encodings (e.g., TF-IDF); (2) larger LLMs yield consistent gains; (3) refining LLMs on KGs provides modest but reliable improvements in performance and parameter efficiency; (4) when entity linking is solved, KG embeddings perform on par with same-sized LLMs, suggesting that LLMs’ main advantage lies in implicit symbol grounding rather than superior knowledge quality.

**Strengths:**

- Comprehensive and well-structured benchmark: The collection of 105 diverse datasets from three distinct sources (TextTabBench, CARTE, WikiDBs) provides strong external validity. The construction of a 15-table “entity-linked” subset is a methodological highlight that enables clean isolation of the entity linking factor.

- Clear, actionable insights: The paper convincingly demonstrates that representation quality—not just model architecture—is the bottleneck in text-rich tabular learning. The conclusion that LLMs primarily help by solving symbol grounding is both nuanced and valuable for the community.

- Reproducibility: Experimental protocols (train sizes, random seeds, PCA dimensionality, estimator choices) are thoroughly documented, and runtime costs are reported—enhancing practical utility.

- Balanced model coverage: The evaluation spans a wide spectrum of models, from classic KG embedders (RotatE, ComplEx) to modern LLMs (Llama-3, Qwen3) and hybrid approaches (Knowledge Card, ERNIE).

**Weaknesses:**

- Narrow scope of KG evaluation: Pure KG models are only evaluated on the 15 linked tables, which represent a best-case scenario. The paper does not assess how KG-based methods degrade under realistic, noisy, or partial entity linking—thus overestimating their practical applicability.

- Lack of privacy or robustness considerations: Given the focus on external knowledge, the paper overlooks critical issues such as leakage of sensitive entities via embeddings, vulnerability to adversarial entity perturbations, or compatibility with privacy-preserving learning (e.g., federated or differentially private settings).

- Downstream estimator mismatch: The use of PCA to compress high-dimensional LLM/KG embeddings before feeding them to XGBoost or TabPFNv2 may discard useful structure. The paper does not explore alternative integration strategies (e.g., late fusion, attention-based conditioning).

**Questions:**

- Extend KG experiments to realistic linking scenarios: Include experiments with automatic entity linking (e.g., using BLINK or OpenTapioca) and report performance as a function of linking accuracy. This would bridge the gap between idealized and real-world deployment.

- Discuss privacy and security implications: Even a brief discussion of risks (e.g., membership inference from KG-enhanced embeddings) would align the work with contemporary concerns in data-centric AI.

- Explore alternative integration mechanisms: Beyond simple embedding + linear model, consider lightweight adapters or cross-attention modules that preserve the geometry of knowledge-rich representations when used with tabular foundation models.

- Clarify the role of column context: While row serialization includes column names, the ablation on contextual disambiguation (e.g., “Cambridge, UK” vs. “Cambridge, MA”) is only mentioned qualitatively. A quantitative analysis would strengthen the claim.

---

> ### Author Response · Authors · 2025-11-27
> **Response to reviewer QSCG**
>
> We sincerely thank the reviewer for the positive and insightful feedback. We are glad that the review highlights the comprehensiveness and usefulness of our work.
>
> Below, we address the specific questions raised in the review. Corresponding updates have been integrated to the paper (written in blue).
>
> **Automatic and realistic entity linking**
>
> Following the suggestion, we conducted an additional experiment in which table entries are automatically linked to Wikidata entities using BLINK. The results, now reported in Sec. 4.4 (Fig. 5), show that **under this noisy and partial linking scenario KG-based representations degrade substantially**, while LLM-derived embeddings consistently outperform them. This underscores the **practical advantage of LLMs,** which implicitly perform entity resolution.
>
> **Column-context ablation**
>
> Our initial row serialization strategy followed prior work (TabLLM, TabuLa), which reported benefits of including column names. To complement our qualitative motivation, we conducted an experiment removing column names from the serialization. Results (Appendix B.2, Fig. 12, p. 18) show that, **on average, including column context provides a small but statistically significant performance improvement** (t-test).
>
> We observe variation across datasets: for some tables, column context helps, while for others it does not. Qualitative analysis in Appendix C.2 suggests that **column context is less helpful, or even slightly detrimental, when headers are generic or non-informative.**
>
> **Alternative integration mechanisms**
>
> Exploring adapters or attention-based modules to better preserve high-dimensional embedding structure is interesting but would require a dedicated training paradigm, which is beyond the scope of our current study.
>
> To address concerns about PCA, we conducted an additional experiment comparing Ridge regression with and without PCA at different dimensions. Results (Sec.5.1, Fig. 8a) show that PCA only slightly affects performance, indicating that the **compression does not significantly degrade the utility of the embeddings, and even improves them for TabPFN.** While more sophisticated integration strategies remain a promising direction for future work, our current results suggest that PCA provides a reasonable and practical compromise.
>
> **Privacy and security implications**
>
> While our current study focuses on benchmarking knowledge-rich embeddings for tabular learning, we acknowledge that external knowledge sources introduce potential privacy and robustness concerns. For instance:
> - LLM or KG embeddings could, in principle, encode sensitive information, raising the risk of membership inference or unintended leakage.
> - Adversarial perturbations to table entries or entity links could degrade downstream performance or mislead models.
> - Integration with privacy-preserving paradigms (e.g., federated learning or differential privacy) is not explored in this work.
>
> We added a discussion of these issues in the revised version of the paper, highlighting them as important considerations for safe deployment and directions for future research in the final discussion.

---

### Official Review · Reviewer_wUU1 · 2025-10-31

**Soundness:** 3
**Presentation:** 3
**Contribution:** 2
**Rating:** 2
**Confidence:** 4

**Summary:**

This paper investigates how external knowledge can enhance tabular learning, particularly when tables include textual fields referencing real-world entities. The authors systematically compare two major sources of background knowledge:
(1) LLMs that implicitly encode extensive factual and semantic information; and
(2) Knowledge Graphs (KGs) that provide explicit, curated relational structures but depend on entity linking.
To enable a controlled comparison, the paper introduces a benchmark of 105 tabular datasets (drawing from TextTabBench, CARTE, and WikiDBs), encompassing both classification and regression tasks. A wide range of representation strategies are evaluated, e.g. non-pretrained encoders, LLM-based embeddings, KG embeddings, and hybrid LLM+KG refinements—paired with several tabular predictors, including ridge regression, XGBoost, and TabPFNv2. The study reveals several key findings.

**Strengths:**

- The work addresses an underexplored yet practically important research question—how to inject background knowledge into tabular learning—bridging symbolic reasoning (KGs) and neural representation learning (LLMs).
- The experiments cover a broad spectrum of models, from lightweight text encoders to multi-billion parameter LLMs, as well as multiple downstream learners. This comprehensive setup strengthens the credibility of the conclusions.
- The empirical observation that LLMs and KGs converge in performance after entity linking offers an interesting theoretical insight into how implicit and explicit factual knowledge may complement each other.

**Weaknesses:**

- The study mainly explores feature-level embeddings followed by downstream predictors. It omits more advanced fusion methods (e.g., cross-attention, joint training, or representation alignment) that could yield richer interactions between tabular and knowledge-based features.
- The finding that ridge regression outperforms more complex learners may stem from dimensionality reduction artifacts (e.g., PCA bottlenecks), potentially underestimating the capabilities of non-linear models like XGBoost and TabPFNv2.
- The paper reports aggregate metrics but does not provide qualitative case studies to illustrate where LLMs or KGs perform particularly well or poorly (e.g., domain-specific tables, rare entities, or ambiguous text).
- The definition of “refinement” is inconsistent across baselines (ERNIE, Knowledge Card, KGT5). Without further controlled ablations isolating architecture, scale, and pretraining data, it remains unclear what drives the observed gains.
- Some Wikipedia-derived datasets may resemble document classification rather than genuinely heterogeneous tabular learning, weakening claims about handling tabular structure.
- The paper could better position itself relative to: 1) Retrieval-Augmented Generation (RAG) approaches that dynamically incorporate knowledge. 2) Multimodal table encoders such as TaBERT, TURL, and TAPAS, which explicitly integrate table structure and text.

**Questions:**

- Could retrieval-based approaches (e.g., RAG-style KG or LLM lookups) outperform static embeddings while retaining interpretability?
- How does the method behave under noisy or partial entity linking? Could uncertainty in linking be explicitly modeled?
- Have the author(s) considered multi-column or relational dependencies, such as type hierarchies or foreign-key relationships, beyond row-level concatenation?
- Would fine-tuning smaller models with knowledge-based pretraining narrow the performance gap with large LLMs?
- Could benchmarking against retrieval-based tabular models (e.g., RAG-TAB, KnowTab) provide deeper insight into dynamic vs. static knowledge integration?
- Why does ridge regression outperform TabPFNv2 after embedding projection—does PCA distort representation geometry or reduce model flexibility?
- Are the performance improvements primarily due to semantic enrichment (better factual knowledge) or dimensional expansion (higher embedding capacity)?


I would consider raising my score if the authors can adequately address these questions.

---

> ### Author Response · Authors · 2025-11-27
> **Response to reviewer wUU1**
>
> We thank the reviewer for the detailed review. Below, we provide a response to the reviewer’s questions.
>
> **Comparison of KGs and LLMs under noisy or partial entity linking**
>
> We agree that examining robustness to imperfect entity linking is important for practical use of KG embeddings. Since there are no established tools for linking tabular cell values to KG entities, we adapted BLINK, a state-of-the-art linker for textual mentions. We applied BLINK to serialized table rows to obtain links to Wikipedia entities, and then integrated the corresponding Wikidata5M embeddings into the table (details in Appendix A.4, pp. 17–18).
>
> As shown in Sec. 4.4 (Fig. 5, p. 8), under this noisy and partial linking scenario KG embeddings degrade substantially, and LLM-based representations clearly outperform them. This suggests that **the implicit entity-resolving ability of LLMs provides a significant practical advantage** when high-quality linking cannot be guaranteed.
>
> **Effect of dimensionality reduction**
>
> We thank the reviewer for this point. To clarify the role of dimensionality reduction, we conducted an additional experiment where all downstream learners (Ridge, XGBoost, TabPFNv2) were trained on the **same** PCA-reduced embeddings (d = 300). For Ridge and TabPFNv2, we also compare these results with a PCA of d=500. The results show that:
> - **Knowledge-rich representations continue to outperform non-pretrained baselines even after PCA** (Sec. 5.1 fig 7), indicating that gains come primarily from richer semantic content rather than from higher dimensionality.
> - **TabPFNv2 performs better after more compression** (from d = 500 to d = 300), despite the information loss. This highlights TabPFNv2’s difficulty in handling very high-dimensional knowledge-rich embeddings (Sec. 5.2 fig 8). This is unlike Ridge, where PCA slightly reduces the performance, as expected from information loss.
> - **Ridge still outperforms XGBoost and TabPFNv2** when all models use identical reduced vectors (Sec. 5.1 fig 7). This suggests that the geometry of the extracted embeddings aligns better with linear inductive biases, whereas current tabular learners struggle to exploit these features effectively.
>
> **Qualitative case studies**
>
> We added several case studies in Appendix C (p. 19). Tables 11–12 (pp. 26–27) provide examples where LLM- or KG-based representations perform particularly well, and Tables 13–14 (p. 27) analyze when column-level context helps or not. Overall, **LLMs tend to perform best on free-text** or open-domain attributes, whereas **KG embeddings excel in knowledge-intensive or entity-centric columns.** These examples complement our aggregate results and clarify the conditions under which each approach is advantageous.
>
> **Positioning relative to retrieval-based approaches**
>
> We thank the reviewer for raising this point. We expanded the related-work section to more clearly situate our work with respect to retrieval-augmented methods (lines 182-184). To the best of our knowledge (though we failed to find the mentioned baselines, RAG-TAB, KnowTab), existing RAG-style approaches are designed primarily for table QA and text generation, rather than supervised tabular learning pipelines studied here. **Applying RAG in our setting would require defining and indexing a large external corpus, performing retrieval at training and inference time, and managing substantially higher computational cost.** Building such a system is beyond the scope of this paper, but we agree it represents an interesting direction for future work.
>
> Importantly, our experiments show that static knowledge-rich representations (LLM- or KG-based) already provide strong benefits while remaining lightweight and easy to integrate into standard tabular learners. Whether dynamic retrieval could further improve performance, while retaining interpretability and tractability, is an open question that we now highlight in our conclusion (line 526).
>
> **Fine-tuning smaller models**
>
> Sec. 4.3 studies this question by comparing knowledge-fine-tuned LLMs to their base counterparts. As shown in Fig. 4, a regression of performance against model size indicates that **knowledge-refined models achieve accuracy comparable to that of purely pretrained LLMs roughly 1.5 times larger.** This suggests that targeted knowledge pretraining can indeed narrow the gap between small and large models, providing a more compute-efficient alternative to scaling.
>
> **Multi-column or relational dependencies**
>
> We agree that incorporating relational signals, such as type hierarchies or foreign-key links, would be valuable. However, these signals are highly dataset- and schema-specific and typically require domain expertise or manual engineering. Since our goal is to evaluate general-purpose text encoders in a fully automated, schema-agnostic setting, we restrict our study to row-level representations. Extending the framework to richer relational structures is an interesting direction for future work.

---

### Official Review · Reviewer_PSi8 · 2025-11-01

**Soundness:** 3
**Presentation:** 3
**Contribution:** 2
**Rating:** 6
**Confidence:** 4

**Summary:**

This paper investigates how external background knowledge can be integrated into tabular learning, focusing on textual columns that reference real-world entities (e.g., drugs, companies, locations).
The authors benchmark 105 text-containing tabular datasets and compare representations derived from **large language models (LLMs)**, **knowledge graphs (KGs)**, and **hybrid LLM + KG models**.
They find that knowledge-rich representations substantially improve downstream prediction—often more than using sophisticated tabular learners—and that larger LLMs provide greater gains. Refining LLMs on KGs improves parameter efficiency, and in the idealized case where all entities are perfectly linked, KG embeddings perform on par with LLMs.
The study concludes that combining LLMs and KGs is a promising direction for future tabular foundation models.

**Strengths:**

* The paper tackles a novel and meaningful problem—bridging tabular learning and external knowledge.
* Comprehensive empirical study across 105 datasets, with diverse textual attributes.
* Systematic comparison of LLM, KG, and hybrid models; clear identification of the entity-linking bottleneck.
* Results suggest interesting scaling trends and show that “representation quality > model complexity” in importance.

**Weaknesses:**

* **Over-restrictive assumptions:** removing all numerical features and focusing solely on text columns creates an artificial setting; results may not generalize to realistic multi-modal tables.
* **Limited real-world relevance:** experiments are confined to small-data regimes (64 / 256 / 1024 samples), which are uncommon in industrial tabular tasks.
* **Lack of raw quantitative results:** only normalized gains are reported; absolute AUC/R² improvements may be modest.
* **No discussion of KG construction or cost:** while KG embeddings are used, the paper does not analyze the effort required for entity linking or graph maintenance, undermining claims of practical benefit.
* **Reproducibility issues:** key configuration details (exact sampling splits, variance across seeds, hyper-parameters for embedding extraction) are only briefly mentioned.

**Questions:**

1. Since the same test sets are used across training sizes {64, 256, 1024}, how does performance evolve with more training samples? Do the relative advantages of LLM / KG embeddings diminish as data grows?
2. Could the authors release or at least summarize the **raw experimental tables** (AUC/R² per dataset) to enhance reproducibility and allow independent meta-analysis?
3. A valuable extension would be to re-introduce **numerical features** and study how numerical and textual features interact—are their contributions orthogonal or redundant? This would make the findings more applicable to real-world tabular pipelines.

---

> ### Author Response · Authors · 2025-11-27
> **Response to reviewer PSi8**
>
> We thank the reviewer for their thoughtful and positive assessment. We are glad that the reviewer recognizes both the novelty and comprehensiveness of our study.
>
> We address the reviewer’s questions below and have updated the manuscript accordingly (revisions are in blue).
>
> **Q1: Effect of training size**
>
> We thank the reviewer for raising this important point. We initially limited our study to 1,024 samples since several of our (real-world) datasets do not go beyond this size. However, to assess whether our conclusions hold beyond the small-data regime, **we ran additional experiments on a subset of 49 datasets containing more than 10,000 rows**, and plotted learning curves covering the range 64 to 10,000 samples. The results are now included in Appendix B.1 (Fig. 11, p. 18).
>
> We find that while all models benefit from more data, the relative ordering between LLM, KG-refined models, and non-pretrained encoders remains remarkably stable. In particular, knowledge-rich representations (LLM or KG-based) continue to outperform purely surface-form encodings even at 10k samples. Thus, **the advantages we report do not vanish with increasing data**; rather, they persist across training sizes, reinforcing our main conclusions.
>
> **Q2: Reporting raw results**
>
> Our main experiment includes 19 text-encoding models, 3 downstream estimators, 3 training sizes, and 105 datasets, which would require several hundred per-dataset tables if presented in the appendix. We believe this would make the supplementary material difficult to navigate. Nevertheless, we fully agree that providing raw scores is important for reproducibility and for enabling independent meta-analysis.
>
> To address this, **we now commit to releasing the complete set of raw AUC/R2 results** (per dataset, model, estimator, and training size) **as CSV files** in our public GitHub repository upon publication. In addition, **we have added summary tables** reporting non-normalized mean scores per model and estimator, separately for regression and classification, aggregated over each dataset source (Tables 16-18, pp. 28-30, Appendix D.2).
>
> **Q3: Re-introducing numerical features**
>
> We added a new experiment in which we re-introduce the original numerical features alongside textual ones (Sec. 5.2, p. 10; Fig. 9). This allows us to directly assess how numerical and text-derived features interact in heterogeneous tables.
> The results show that:
> - The **relative ranking of text encoders remains unchanged** when numerical features are added.
> - Combining numerical + textual features substantially outperforms either modality alone, indicating that their **contributions are complementary rather than redundant.**
> - Text-only features consistently outperform number-only features, underscoring the **dominant role of textual signal** in our benchmark.
> - Knowledge-rich representations provide large gains for Ridge, but far smaller gains for TabPFNv2, suggesting that current table foundation models struggle to fully leverage high-dimensional knowledge-enhanced embeddings. This is also supported by our new analysis on the effect of PCA (Sec. 5.1).
>
> In appendix B.3, Figure 13, we show that incorporating numerical features alongside text embeddings yields similar improvements across LLMs of different sizes, suggesting that **the information captured by richer models is complementary to, rather than redundant with, numerical features.**
>
> We sincerely thank the reviewer for raising this interesting question. We have updated the conclusion to reflect these findings and to better situate our results in the context of tables with mixed data types.
>
> **Discussion of KG cost**
>
> We have expanded the paper to make the practical costs of KG-based methods more explicit. In particular, we have added an experiment using automatic entity linking with BLINK (Sec. 4.4, Fig. 5). **The computational cost of linking is reported in Appendix A.4** (Table 7, p23). BLINK requires ~30 minutes per dataset on average, whereas LLM-based embedding extraction takes from a few seconds to ~2 minutes depending on model size (Table 15, p 28).
>
> KG models are cheaper to train but expensive to use due to the entity-linking step, while LLMs have high pre-training cost but are fast at inference. Regarding construction and maintenance, private KGs are expensive, but for public KGs like Wikidata the costs are amortized among many users.

---

### Author Response · Authors · 2025-12-02
**Summary of improvements and key contributions**

# Summary of the improvements
We thank all reviewers for their feedback, which prompted us to make significant improvements to our original submission. Here, we provide a summary of these improvements, visible in blue in the updated PDF.

### **Inclusion of numerical values (PSi8, NUSu) - Section 5.2**

Our initial removal of numerical features aimed to isolate the predictive contribution of text features. As a complementary analysis, **we reintroduced numerical features**, to study how they interact with text features. The results show that:

- The relative **ranking of text encoders remains unchanged**.
- Combining numerical + text features markedly outperforms either alone, showing **complementary predictive value**.
- Text-only features outperform numbers-only, highlighting the **primary importance of text features**.
- Knowledge-rich representations provide huge benefits for Ridge, but little to TabPFNv2, highlighting the **shortcomings of table foundation models for fully leveraging these representations**.

### **Clarify the impact of PCA (wUU1, QSCG) - Section 5.1**

Our initial approach was to use each downstream estimator to the best of its ability, thus using PCA only when necessary (so for XGBoost and TabPFNv2, but not Ridge). We added experiments evaluating Ridge, XGBoost, and TabPFNv2 on the same PCA-reduced embeddings, showing that **our conclusion that Ridge is better-suited for knowledge-rich embeddings still holds once the confounding factor of PCA is removed**.

This experiment has also highlighted a **key limitation of TabPFNv2**, that we now emphasize in our conclusion: it is **unable to leverage high-dimensional, knowledge-rich embeddings** since its performance improves when compressed from 500 to 300 dimensions, despite the information loss.

### **Noisy/partial entity linking (wUU1, QSCG) - Section 4.4, & Appendix A.4**

To examine robustness of KG embeddings to imperfect entity linking, we used a SOTA linker for textual mentions, BLINK. We adapted it to serialized table rows to obtain links between table entries and Wikidata entities (see Appendix A.4).

The results show that, under imperfect linking, LLM-based representations clearly outperform KG embeddings. This suggests that **the implicit entity-resolving ability of LLMs provides a significant practical advantage** when high-quality linking cannot be guaranteed.

### **Modern tabular baselines (NUSu) - Appendix D.3**

We added the fine-tuning version of TARTE (TARTE-FT), a successor to CARTE and a recent SOTA end-to-end model for tables with mixed data types. Evaluated on tables with both numerical and text features, the results further reinforced our conclusion: for tabular learning with text, **the largest gains come from knowledge-rich text representations, rather than architectural sophistication alone**, highlighting the need for future table foundation models that leverage LLM-based text representations.

### **Other reviewers’ concerns**

In addition, we made the following improvements:
- Adding larger train-sizes up to 10k (PSi8, NUSu): Appendix B.1 (Fig. 11)
- Reporting raw results (PSi8): Appendix D.2 (Tables 16-18)
- Quantifying the role of column context (QSCG): Appendix B.2 (Fig. 12)
- Qualitative case studies (wUU1): Appendix C (Tables 11-14)
- Positioning relative to retrieval-based methods (wUU1): Section 2 (182-184) and Section 6 (526-527)
- Discussing privacy/security implications (QSCG): Section 6 (516-517)
- Discussing KG cost (PSi8): Appendix A.4 (Table 7)

# Summary of our contributions

Tabular data with text is a central yet understudied topic for real-world applications. Our **large-scale, systematic benchmark across 105 diverse datasets** addresses open questions like: *How should text in tables be encoded? Should future table foundation models be pretrained on KGs, free text, or both? And where do current models fall short?*

Our paper supports the following key messages:
- **Knowledge-rich text representations**, either from KGs or LLMs, can **substantially improve the quality of the predictions**.
- **But current table foundation models fail to leverage these representations**. They struggle with high-dimensional inputs, revealing a fundamental limitation and a clear opportunity for future research.
- **Large, knowledge-enhanced LLMs offer the most promising path forward.** Larger LLMs consistently perform best. Their implicit solving of the symbol grounding problem gives them an edge over pure KG solutions. However refining these LLMs on knowledge can provide additional gains, pointing towards a promising synergy: pretraining LLM-based tabular models on large knowledge bases, such as Wikidata.

We believe the clarified insights and strengthened empirical evidence will help **guide future research on table foundation models to enable them exploiting text features to their full potential**. Given the ubiquity of tables containing text, we are convinced of the **high practical impact** of this work.

---

### Meta-Review · Area_Chair_cy75 · 2026-01-04

**Summary:**

This paper studies how external background knowledge can be leveraged for tabular learning with textual columns, comparing representations from LLMs, KGs, and hybrid LLM+KG models across benchmark datasets. One of the key findings is that knowledge-rich representations substantially improve downstream prediction—often more than using sophisticated tabular learners—and that larger LLMs provide greater gains. Details about the reviewers' major concerns are described below.

**Reviewer Concerns:**

*Concerns largely addressed by the rebuttal:

- Artificial text-only setting (PSi8, NUSu): New experiments re-introducing numerical features are provided.

- Small-data regime and generalization (PSi8, NUSu): Added learning curves up to 10k samples.

- Noisy entity linking (QSCG, wUU1): Automatic BLINK-based linking is discussed.

- Effect of PCA (wUU1, QSCG): Additional PCA ablations show that gains are driven by semantic enrichment rather than dimensionality alone.

*Concerns still outstanding:

- Limited exploration of advanced fusion or joint-training methods (wUU1, QSCG): The authors did not explicitly discuss this point.

- Dependence on static embeddings and row-level representations: Multi-column relational modeling and dynamic retrieval remain future work.

- Incremental nature of LLM+KG refinement (NUSu): While validated at scale, the contribution is more confirmatory than algorithmically novel.

**Reviewer Scores:**

While some reviewers might have adjusted the scores slightly if they had been able to participate fully in the discussion, the overall decision would have likely leaned toward rejection due to the major remaining concerns described above.

---

### Decision · Program_Chairs · 2026-01-26

Reject